# 4DBInfer: A 4D Benchmarking Toolbox for Graph-Centric Predictive Modeling on RDBs

**Minjie Wang**[1*]**, Quan Gan**[1*]**, David Wipf**[1]**, Zhenkun Cai**[1]**, Ning Li**[2†]**, Jianheng Tang**[3†]**,
Yanlin Zhang**[4†]**, Zizhao Zhang**[5†]**, Zunyao Mao**[6†]**, Yakun Song**[2†]**, Yanbo Wang**[7†]**, Jiahang Li**[8†]**,
Han Zhang**[2†]**, Guang Yang**[1]**, Xiao Qin**[1]**, Chuan Lei**[1]**, Muhan Zhang**[7]**, Weinan Zhang**[2]**,
Christos Faloutsos**[1,9]**, Zheng Zhang**[1]

[1]Amazon  [2]Shanghai Jiao Tong University  [3]Hong Kong University of Science and Technology
[4]Fudan University  [5]Tsinghua University  [6]Southern University of Science and Technology
[7]Peking University  [8]Hong Kong Polytechnic University  [9]Carnegie Mellon University

## Abstract

Given a relational database (RDB), how can we predict missing column values in some target table of interest? Although RDBs store vast amounts of rich, informative data spread across interconnected tables, the progress of predictive machine learning models as applied to such tasks arguably falls well behind advances in other domains such as computer vision or natural language processing. This deficit stems, at least in part, from the lack of established/public RDB benchmarks as needed for training and evaluation purposes. As a result, related model development thus far often defaults to tabular approaches trained on ubiquitous single-table benchmarks, or on the relational side, graph-based alternatives such as GNNs applied to a completely different set of graph datasets devoid of tabular characteristics. To more precisely target RDBs lying at the nexus of these two complementary regimes, we explore a broad class of baseline models predicated on: (i) converting multi-table datasets into graphs using various strategies equipped with efficient subsampling, while preserving tabular characteristics; and (ii) trainable models with well-matched inductive biases that output predictions based on these input subgraphs. Then, to address the dearth of suitable public benchmarks and reduce siloed comparisons, we assemble a diverse collection of (i) large-scale RDB datasets and (ii) coincident predictive tasks. From a delivery standpoint, we operationalize the above *four dimensions* (4D) of exploration within a unified, scalable open-source toolbox called *4DBInfer*; please see
`https://github.com/awslabs/multi-table-benchmark/`.

## 1 Introduction

Relational databases (RDBs) can be viewed as storing a collection of interrelated data spread across multiple linked tables. Of vast and steadily growing importance, the market for RDB management systems alone is expected to exceed $133 billion USD by 2028 [56]. Even so, while the machine learning community has devoted considerable attention to predictive tasks involving *single* tables, or so-called tabular modeling tasks [21, 49, 58], thus far efforts to widen the scope to handle *multiple* tables and RDBs still lags behind, despite the seemingly enormous potential of doing so. With respect to the latter, in many real-world scenarios critical features needed for accurately modeling a

---

*Equal contribution; correspondence to {minjiw,quagan}@amazon.com.
†Work done during internship at Amazon.

38th Conference on Neural Information Processing Systems (NeurIPS 2024) Track on Datasets and Benchmarks.

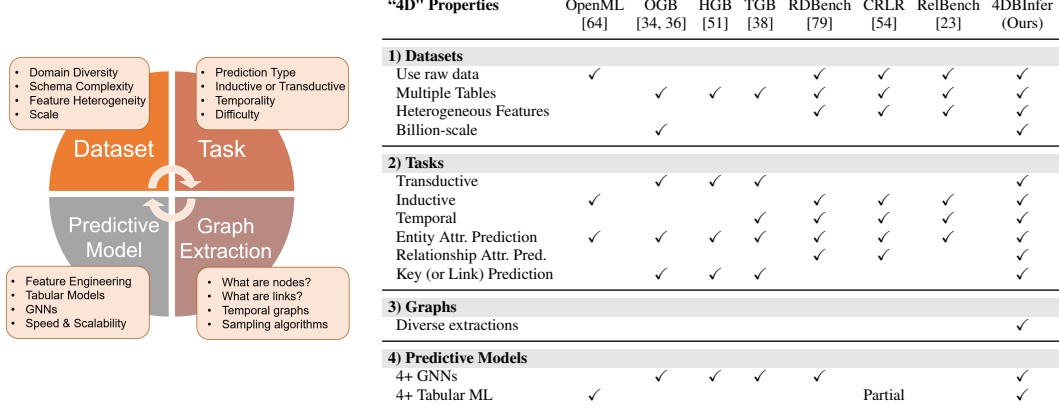

| "4D" Properties | OpenML [64] | OGB [34, 36] | HGB [51] | TGB [38] | RDBench [79] | CRLR [54] | RelBench [23] | 4DBInfer (Ours) |
|---|---|---|---|---|---|---|---|---|
| **1) Datasets** | | | | | | | | |
| Use raw data | ✓ | | | | ✓ | ✓ | ✓ | ✓ |
| Multiple Tables | | ✓ | ✓ | ✓ | ✓ | ✓ | ✓ | ✓ |
| Heterogeneous Features | | | | | ✓ | ✓ | ✓ | ✓ |
| Billion-scale | | ✓ | | | | | | ✓ |
| **2) Tasks** | | | | | | | | |
| Transductive | | ✓ | ✓ | ✓ | | | | ✓ |
| Inductive | ✓ | | | | ✓ | ✓ | ✓ | ✓ |
| Temporal | | | | ✓ | ✓ | ✓ | ✓ | ✓ |
| Entity Attr. Prediction | ✓ | ✓ | ✓ | ✓ | ✓ | ✓ | ✓ | ✓ |
| Relationship Attr. Pred. | | | | | ✓ | ✓ | | ✓ |
| Key (or Link) Prediction | | ✓ | ✓ | ✓ | | | | ✓ |
| **3) Graphs** | | | | | | | | |
| Diverse extractions | | | | | | | | ✓ |
| **4) Predictive Models** | | | | | | | | |
| 4+ GNNs | | ✓ | ✓ | ✓ | ✓ | | | ✓ |
| 4+ Tabular ML | ✓ | | | | Partial | | | ✓ |

Figure 1: 4DBInfer exploration dimensions. Unlike prior benchmarking efforts (table columns on right), 4DBInfer considers an evaluation space with diversity across the 4D Cartesian product of (i) datasets, (ii) tasks, (iii) graph extractors, and (iv) predictive baselines. See Sections 3 and 4 (and in particular Section 4.2) for further details of table properties and assumptions.

given quantity of interest are not constrained to a single table [9, 14], nor can be easily flattened into a single table via reliable/obvious feature engineering [15].

This disconnect between commercial opportunity and academic research focus can, at least in large part, be traced back to one transparent culprit: Unlike widely-studied computer vision [16], natural language processing [67], tabular [28], and graph [35] domains, established benchmarks for evaluating predictive ML models of RDB data are much less prevalent. This reality is an unsurprising consequence of privacy concerns and the typical storage of RDBs on servers with heavily restrictive access and/or licensing protections. With few exceptions (that will be discussed in later sections), relevant model development is instead predicated on surrogate benchmarks that branch as follows.

Along the first branch, sophisticated models that explicitly account for relational information are often framed as graph learning problems, addressable by graph neural networks (GNNs) [6, 29, 32, 37, 42, 45, 57, 66] or their precursors [78, 80, 81], and evaluated specifically on graph benchmarks [35, 43, 51]. The limitation here though is that performance is conditional on a fixed, pre-specified graph and attendant node/edge features intrinsic to the benchmark, not an actual RDB or native multi-table format. Hence the inductive biases that might otherwise lead to optimal performance on the original data can be partially masked by whatever process was used to produce the provided graphs and features. As for the second branch, emphasis is placed on tabular model evaluations that preserve the original format of single table data, possibly with augmentations collected from auxiliary tables. But here feature engineering and table flattening are typically prioritized over exploiting rich network effects as with GNNs [9, 14, 47, 48]. Critically though, currently-available head-to-head comparisons involving diverse candidate approaches representative of *both* branches on un-filtered RDB/multi-table data are insufficient for drawing clear-cut conclusions regarding which might be preferable and under what particular circumstances.

To address the aforementioned limitations and help advance predictive modeling over RDB data, in Section 2 we first introduce a generic supervised learning formulation across both inductive and transductive settings covering dynamic RDBs as commonly-encountered in practice. A given predictive pipeline is then specified by (i) a sampling/distillation operator which extracts information most relevant to each target label, followed by (ii) a trainable prediction model. In Section 3 we present a specific design space for these two components. For the former, we adopt a graph-centric perspective whereby distillation is achieved (either implicitly or explicitly) via graphs and sampled subgraphs extracted from RDBs. Meanwhile, for the latter we incorporate trainable architectures that represent strong exemplars drawn from *both* tabular and graph ML domains. We emphasize here that until more extensive benchmarking has been conducted, it is advisable not to prematurely exclude candidates from either domain, or hybrid combinations thereof. In this regard, Section 4 introduces a new suite of RDB benchmarks along with discussion of the comprehensive desiderata which leads to them. These include multiple diversity/coverage considerations across both (i) datasets and (ii) predictive tasks, while also resolving limitations of existing alternatives. Our 4DBInfer toolbox for

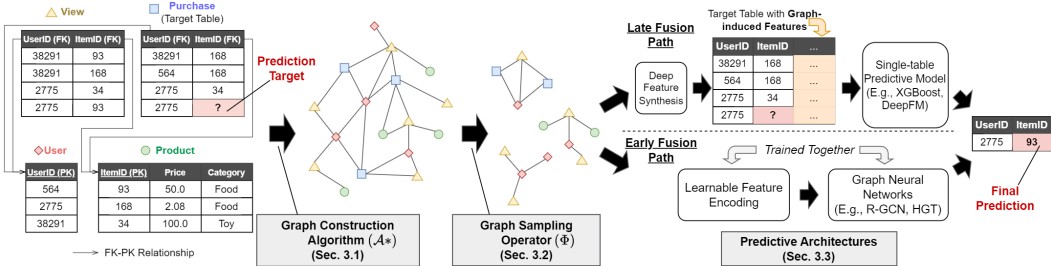

Figure 2: 4DBInfer overview. *Left*: First a (i) RDB dataset and (ii) task (i.e., predictive target here) are selected from among proposed benchmarks. *Middle*: Then a (iii) graph extractor/sampling operator is chosen which converts the RDB and task into subgraph chunks (*middle*). *Right*: Lastly a (iv) predictive model ingests these chunks, either through early or late feature fusion, to produce an estimate of the target values (*right*).

pairing a so-called *2D* design space of baseline models from Section 3 and the *2D* benchmark coverage from Section 4 within a neutral combined *4D* evaluation setting is introduced in Section 5. And finally, Section 6 culminates with representative experiments conducted using 4DBInfer.

In tracing these endeavors, our paper consolidates the following contributions:

- **2D Space of Baselines:** On the *modeling side* we describe a *2D* design space with considerable variation in (i) graph construction/sampling operators and (ii) trainable predictor designs. The latter covers popular choices drawn from GNN and tabular domains, representative of both early and late feature fusion strategies. This diversity safeguards against siloed comparisons between pipelines of only a single genre, e.g., tabular, GNNs.
- **2D Space of Benchmarks:** On the *data side*, we introduce a *2D* suite of RDB benchmarking (i) datasets and (ii) tasks that are devoid of potentially lossy or confounding pre-processing that might otherwise skew performance in favor of one model class or another. These benchmarks also vary across key dimensions of scale (e.g., up to 2B RDB rows), source domain, RDB schema, and temporal structure.
- **4DBInfer Toolbox:** We operationalize the above via a *unified and scalable open-source toolbox* called 4DBInfer that facilitates direct head-to-head empirical comparisons across each dimension of baseline model and benchmarking task (and is readily extensible to accommodate new additions of either). Figure 1 depicts the combined *4D* exploration space of 4DBInfer, along with comparisons relative to existing RDB, tabular, and graph benchmarking work.
- **Empirical Support:** Experiments using 4DBInfer highlight the relevance of each of the proposed four dimensions of exploration to the design of successful RDB predictive models, as well as the limitations of more naive commonplace approaches such as simply joining adjacent tables.

## 2 Predictive Modeling on RDBs

An RDB $\mathcal{D}$ [25] can be viewed as a set of $K$ tables denoted as $\mathcal{D} := \{\boldsymbol{T}^k\}_{k=1}^K$, where $\boldsymbol{T}^k$ refers to the $k$-th constituent table defined by a particular entity type. Each row of a table then represents an instance of the corresponding entity type (e.g., a user), while the columns contain relevant features of each such instance (e.g., user profile information). Such features are typically heterogeneous and may include real values, integers, categorical variables, text snippets, or time stamps among other things. We adopt $\boldsymbol{T}_{i:}^k$ and $\boldsymbol{T}_{:j}^k$ to reference the $i$-the row and $j$-th column of $\boldsymbol{T}^k$ respectively. What establishes $\mathcal{D}$ as a *relational* database, as opposed to merely a collection of tables, is that certain table columns are designated as either *primary keys* (PKs) or *foreign keys* (FKs). A column $\boldsymbol{T}_{:j}^k$ serves as a PK when each element is a unique index referencing a row of $\boldsymbol{T}^k$, such as a user ID for example. In contrast, $\boldsymbol{T}_{:j}^k$ is defined as a FK column if each $\boldsymbol{T}_{ij}^k$ corresponds with a unique PK value referencing a row in *another* table $\boldsymbol{T}^{k'}$ (generally $k' \neq k$, although this need not strictly be the case), with the only restriction being that all such indices within a given FK column must point to rows within the same table. In this way, the domain of any FK column is given by the corresponding PK column it references. Please see l.h.s. of Figure 2 for a simple RDB example.

## 2.1 Making Predictions over Dynamic RDBs

Generally speaking, RDBs are *dynamic*, with information regularly being added to or removed from $\mathcal{D}$. Hence if we are to precisely define a predictive task involving an RDB, and particularly an inductive task, it is critical that we specify the RDB state during which a given prediction is to occur. For this reason, we refine our original RDB definition as $\mathcal{D}(s) := \{\boldsymbol{T}^k(s)\}_{k=1}^K$, where $s \in \mathcal{S}$ defines the RDB *state* drawn from some set $\mathcal{S}$. Note that $\mathcal{S}$ could simply reflect counting indices (versions) such as the set of natural numbers; importantly though, each $s \in \mathcal{S}$ need not necessarily correspond with physical/real-world time per se, even if in some cases it may be convenient to assume so. This then leads to the following core objective:

> **Problem Statement:** *Using all relevant information available in $\mathcal{D}(s)$, predict an unknown RDB quantity of interest $\boldsymbol{T}_{ij}^k(s)$ as uniquely specified by the tuple $\{s, k, i, j\}$, where $s$ determines the state, $k$ the table, and $\{i, j\}$ the table cell we wish to estimate.*

To illustrate, the unknown $\boldsymbol{T}_{ij}^k(s)$ is represented by '?' on the l.h.s. of Figure 2. Ideally, we would like to closely approximate the distribution $p\left(\boldsymbol{T}_{ij}^k(s) \mid \mathcal{D}(s) \setminus \boldsymbol{T}_{ij}^k(s)\right)$, meaning all other information in the RDB is fair game as conditioning variables governing our prediction at state $s$ of missing value $\boldsymbol{T}_{ij}^k(s)$. Of course in practice it is neither feasible nor necessary to condition on the *entire* RDB given limited computational resources and the likely irrelevance of much of the stored data w.r.t. $\boldsymbol{T}_{ij}^k(s)$. Hence our revised objective is to incorporate a sampling operator $\Phi$ defined such that

$$p\left(\boldsymbol{T}_{ij}^k(s) \mid \Phi\left[\mathcal{D}(s) \setminus \boldsymbol{T}_{ij}^k(s)\right]\right) \approx p\left(\boldsymbol{T}_{ij}^k(s) \mid \mathcal{D}(s) \setminus \boldsymbol{T}_{ij}^k(s)\right), \tag{1}$$

where $\Phi\left[\mathcal{D}(s) \setminus \boldsymbol{T}_{ij}^k(s)\right]$ represents a distillation of appreciable information in the RDB relevant to $\boldsymbol{T}_{ij}^k(s)$. As a simple illustrative example, if

$$\Phi\left[\mathcal{D}(s) \setminus \boldsymbol{T}_{ij}^k(s)\right] = \boldsymbol{T}_{i:}^k(s) \setminus \boldsymbol{T}_{ij}^k(s), \tag{2}$$

then all information in $\mathcal{D}(s)$ excluding the features in row $i$ of table $k$ are ignored when predicting $\boldsymbol{T}_{ij}^k(s)$ and we recover a canonical tabular prediction task involving just a single table [21, 49, 58]. More broadly though, $\Phi$ may be defined to select other rows of $\boldsymbol{T}^k(s)$ (i.e., row $i' \neq i$ as used in recent cross-row tabular predictive models [20, 46, 60]), as well as information from other tables $\boldsymbol{T}^{k'}(s)$ (with $k' \neq k$) that are linked to $\boldsymbol{T}^k(s)$ through one or more FK relationships. Even other values in column $\boldsymbol{T}_{:j}^k(s)$ can be incorporated when available, noting that a special case of this scenario can be used to rederive trainable variants of label propagation predictors [70].

## 2.2 High-Level Training and Inference Specs

We now describe training and inference in general terms under an inductive setup; the transductive case will trivially follow as a special case discussed below. We assume target table $k$ and target column $j$ are fixed to define a given predictive task. As such, each training instance is specified by only the tuple $\{s, i\}$, noting that target table row $i$ will often be a function of $s$ by design, e.g., as $s$ increments forward, additional rows with missing values for column $j$ may be added to $\boldsymbol{T}^k(s)$. Let $\mathcal{S}_{tr}$ denote the set of states which have known training labels, and $\psi_{tr}(s)$ the corresponding set of specific indices with labels for each $s \in \mathcal{S}_{tr}$. Then for a given task defined by $k$ and $j$, along with a corresponding sampling operator $\Phi$, we seek to minimize the negative log-likelihood objective

$$\min_{\theta} \sum_{s \in \mathcal{S}_{tr}} \sum_{i \in \psi_{tr}(s)} -\log p\left(\boldsymbol{T}_{ij}^k(s) \mid \Phi\left[\mathcal{D}(s) \setminus \boldsymbol{T}_{ij}^k(s)\right]; \theta\right) \tag{3}$$

with respect to parameters $\theta$ that define the predictive distribution, e.g., a model of the conditional mean for regression problems, or logits for classification tasks, etc. The implicit assumption here is that, when conditioned on $\Phi\left[\mathcal{D}(s) \setminus \boldsymbol{T}_{ij}^k(s)\right]$, each $\boldsymbol{T}_{ij}^k(s)$ is roughly independent of one another

for all $\{\{s, i\} : i \in \psi_{tr}(s), s \in \mathcal{S}_{tr}\}$; this implicit assumption forms the basis of empirical risk minimization [65]. However, it need *not* be the case that individual rows of $\boldsymbol{T}^k(s)$ are unconditionally independent of one another.

Given some $\hat{\theta}$ obtained by minimizing (3), at test time we are presented with new tuples $\{\{s, i\} : i \in \psi_{te}(s), s \in \mathcal{S}_{te}\}$, from which we can compute $p\left(\boldsymbol{T}_{ij}^k(s) \mid \Phi\left[\mathcal{D}(s) \setminus \boldsymbol{T}_{ij}^k(s)\right]; \hat{\theta}\right)$ that ideally approximates the true distribution $p\left(\boldsymbol{T}_{ij}^k(s) \mid \Phi\left[\mathcal{D}(s) \setminus \boldsymbol{T}_{ij}^k(s)\right]\right)$. We remark that a transductive reduction of the above procedure naturally emerges when $s$ is fixed across both training and testing. More generally though, as $s$ increments $\mathcal{D}(s)$ may undergo significant changes, such as new rows appended to $\boldsymbol{T}^k(s)$ (e.g., the 'Purchase' table in Figure 2), new labels/values added to the target column $\boldsymbol{T}_{:j}^k(s)$, as well as arbitrary changes to other tables $\boldsymbol{T}^{k'}(s)$ with $k' \neq k$.

## 3 Design Space of (Graph-Centric) Baseline Models

The general inductive learning framework from the previous section relies on two complementary components: (i) a sampling operator $\Phi$, and (ii) a parameterized predictive distribution as expressed in (3). Collectively, these amount to the first so-called 2D of our proposed 4DBInfer. For both scalability and conceptual reasons, we design the former to operate on graphs that can extracted from RDBs through multiple distinct strategies as summarized in Section 3.1. Subsequently, we will introduce the details of $\Phi$ itself in Section 3.2, followed by choices for predictive architectures in Section 3.3. We also discuss the contextualization with respect to prior work in Appendix H.3.

### 3.1 Converting RDBs to Graphs

A *heterogeneous graph* $\mathcal{G} = \{\mathcal{V}, \mathcal{E}\}$ [63] is defined by sets of node types $V$ and edge types $E$ such that $\mathcal{V} = \bigcup_{v \in V} \mathcal{V}^v$ and $\mathcal{E} = \bigcup_{e \in E} \mathcal{E}^e$, where $\mathcal{V}^v$ references a set of $|\mathcal{V}^v|$ nodes of type $v$, while $\mathcal{E}^e$ indicates a set of $|\mathcal{E}^e|$ edges of type $e$. Both nodes and edges can have associated features. Additionally, any heterogeneous graph can be generalized to depend on a state variable $s$ as $\mathcal{G}(s)$ analogous to $\mathcal{D}(s)$. The goal herein then becomes the establishment of some procedure or mapping $\mathcal{A}^*$ such that $\mathcal{G}(s) = \mathcal{A}^*[(\mathcal{D}(s)]$ for any given RDB of interest.

**Row2Node.** Perhaps the most natural and intuitive way to instantiate $\mathcal{A}^*$ is to simply treat each RDB row as a node, each table as a node type, and each FK-PK pair as a directed edge. Additionally, non-FK/PK column values are converted to node features assigned to the respective rows. Originally proposed in [15] with ongoing application by others [23, 77, 79], we refer to this approach as *Row2Node*.

**Row2N/E.** Importantly though, unlike prior work we do *not* limit 4DBInfer to a single selection for $\mathcal{A}^*$. The motivation for considering alternatives is straightforward: *Even if we believe that graphs are a sensible route for pre-processing RDB data, we should not prematurely commit to only one graph extraction procedure and the coincident downstream inductive biases that will inevitably be introduced.* To this end, as an alternative to Row2Node, we may relax the restriction that every row must be exclusively converted to a node. Instead, rows drawn from tables with more than one FK column can be selectively treated as typed edges, with the remaining non-FK columns designated as edge features. The intuition here is simply that tables with multiple FKs can be viewed as though they were natively a tabular representation of edges. We denote this variant of $\mathcal{A}^*$ as *Row2N/E*.

For full details, analyses, and extensions of both *Row2Node* and *Row2N/E*, please see Appendix I.

### 3.2 Graph-based Sampling Operator $\Phi$

In principle, the sampling operator $\Phi$ need not be explicitly predicated on an extracted graph. However, provided we do not restrict ourselves to *a particular fixed graph upfront*, we are not beholden to any one graph-specific inductive bias. In this way (with some abuse of notation) we instantiate $\Phi$ as

$$\Phi\left[\mathcal{D}(s) \setminus \boldsymbol{T}_{ij}^k(s)\right] \equiv \Phi\left[\mathcal{A}^*\left[\mathcal{D}(s)\right] \setminus \boldsymbol{T}_{ij}^k(s)\right] = \Phi\left[\mathcal{G}(s) \setminus \boldsymbol{T}_{ij}^k(s)\right], \qquad (4)$$

where $\mathcal{A}^*$ is an RDB-to-graph mapping such as described in Section 3.1 and Appendix I, $\mathcal{G}(s) = \mathcal{A}^*[\mathcal{D}(s)]$ represents the extracted graph, and the exclusion operator '$\setminus$' here simply removes the

node feature attribute associated with $\boldsymbol{T}_{ij}^k(s)$ from $\mathcal{G}(s)$. We may now select from among the wide variety of scalable graph sampling methods for finalizing $\Phi$ [7, 10, 32, 73, 76, 83] while specifying the effective receptive field, meaning the number of hops (or tables) away from the target associated with $\boldsymbol{T}_{ij}^k(s)$ from which information is collected. Whatever the choice though, the output of $\Phi$ will be a subgraph of $\mathcal{G}(s)$ containing the target node corresponding to row $\boldsymbol{T}_{i:}^k(s)$.

## 3.3 Trainable Predictive Architectures

At a high-level, once granted $\Phi$ we sub-divide candidate architectures for instantiating the predictive distribution from (3) based on what can be loosely referred to as early versus late feature fusion.

**Late Fusion.** In the context of RDB-specific modeling, we reserve *late fusion* to delineate models whereby parameter-free feature augmentation is adopted to produce a fixed-length, potentially high-dimensional feature vector associated with each target that is, only then, used to train a high-capacity base model with parameters $\theta$ such as those commonly applied to tabular data (this strategy is also referred to as *propositionalization* [47, 75]). For the initial feature augmentation step, we lean on the Deep Feature Synthesis (DFS) framework [41] and extensions thereof for the following reasons:

- DFS is a powerful automated method for generating new features for an RDB by recursively combining data from related tables through aggregation, transformation, etc.
- Although motivated differently, DFS can be re-derived and generalized as a form of subgraph sampling from Section 3.2, followed by concatenated aggregations, as applied to graphs extracted via Row2Node or extensions thereof in Appendix I;
- Special cases of DFS include commonly-used multi-table augmentation and flattening schemes [22], as when paired with sampling limited to 1-hop, or more general multi-hop strategies such as FastProp from the getML package [27];
- DFS can be applied with constraints on $s$ to avoid label leakage;
- DFS is in principle capable of handling large-scale RDBs. Please see Appendices F.1 and G.2 for additional details regarding DFS and our enhanced implementation.

Then for a given target $\boldsymbol{T}_{ij}^k$, this so-called late fusion pipeline produces a fixed-length feature vector

$$\boldsymbol{u}_{ij}^k := \mathrm{DFS}\left[\mathcal{D}(s) \setminus \boldsymbol{T}_{ij}^k(s)\right] \equiv \mathrm{Agg}\left(\Phi\left[\mathcal{G}(s) \setminus \boldsymbol{T}_{ij}^k(s)\right]\right), \tag{5}$$

where Agg is an aggregation operator; see Appendix F.1 for specific choices. And in conjunction with (4), we can subsequently apply any tabular model to estimate the parameters of

$$p\left(\boldsymbol{T}_{ij}^k(s) \mid \Phi\left[\mathcal{D}(s) \setminus \boldsymbol{T}_{ij}^k(s)\right]; \theta\right) \equiv p\left(\boldsymbol{T}_{ij}^k(s) \mid \boldsymbol{u}_{ij}^k; \theta\right) \tag{6}$$

by minimizing (3) over training data. For diversity of tabular base predictors, including both tree- and deep-learning-based, we adopt **MLP**, **DeepFM** [31], **FT-Transformer** [30], **XGBoost** [8], and **AutoGluon (AG)** [3, 21], the latter representing a top-performing AutoML ensembling model. We also remark that DFS combined with gradient boosted trees (akin to our DFS+XGBoost early fusion baseline) has previously been shown to outperform various manual feature engineering and graph embedding methods [14]. We defer broader consideration of benchmarking against handcrafted features to future work.

**Early Fusion.** We next adopt *early fusion* to reference message-passing GNN-like architectures that produce trainable low-dimensional node embeddings (at least relative to late fusion) beginning from the very first model layer. More concretely, for a heterogeneous graph $\mathcal{G}$ (e.g., as extracted from an RDB) these embeddings can be computed as

$$\boldsymbol{h}_{i,\ell}^v = f\left(\left\{\left\{\left(\boldsymbol{h}_{i',\ell-1}^{v'}, vv'\right) : i' \in \mathcal{N}_i^{vv'}\right\} : v' \in \mathcal{N}^v\right\}, \boldsymbol{h}_{i,\ell-1}^v; \theta\right), \tag{7}$$

where $\boldsymbol{h}_{i,\ell}^v$ denotes the embedding of node $i$ of type $v$ at GNN layer $\ell$. In this expression, $\mathcal{N}^v$ indicates the set of node types that neighbor nodes of type $v$, and $\mathcal{N}_i^{vv'}$ is the set of nodes of type $v'$ that neighbor node $i$ of type $v$. Moreover, we assume that there is a unique edge type $e \equiv vv'$ associated with each pair of node types $(v, v')$, as will always be the case for the graphs extracted from RDBs that we focus on here (note also that the edge type $vv'$ is included within the inner-most set definition to differentiate each element within the outer-most set construction). Meanwhile, $f$ is a permutation-invariant function [74] over sets with parameters $\theta$, acting to aggregate or fuse

information from all neighbors of connected node types at each layer. At the output layer, the embeddings produced via (7) can be applied to making node-wise predictions, which translates into predictions of target values in column $T^k_{:j}$.

For implementing $f$ we adopt the popular heterogeneous architectures **R-GCN** [57], **R-GAT** [6], **HGT** [37], and **R-PNA** [13]. Note that we specifically select R-PNA because its core principal neighbor aggregation (extended to heterogeneous graphs) bears considerable similarities to DFS aggregators. In all cases the resulting output layer embeddings will generally depend on which $\mathcal{A}^*$ is used for graph construction. See Appendix F.2 for further details regarding the implementations of $f$.

## 4 A New Suite of RDB Benchmarks

We now introduce and motivate RDB benchmarks that can be applied to evaluating the efficacy of candidate predictive models such as those described in Section 3. This includes a description of our selection desiderata and specific benchmark choices that adhere to them. Please also see Appendix B for a formal definition of an RDB benchmark.

### 4.1 Why New RDB Benchmarks

On the tabular side, there exist countless benchmarks covering every conceivable scenario; however, these are predominately *single-table* datasets, e.g., widely-used Kaggle data [40]. In contrast, on the relational side, benchmarks are often predicated on extracted graphs (usually from limited domains such as citation networks) and pre-processed node features that may have already filtered away useful information [35, 43, 51]. As such, relative performance of candidate models is contingent on what information is available in these graphs and any sub-optimality therein, not actually the original data source. As a simple representative example, on the widely-studied Open Graph Benchmark (OGB) [35], many of the graph datasets were formed from curated citation networks with fixed text embeddings as node features. In this case, researchers have recently found that by reverting back to the original data sources and text features, vastly superior node classification accuracy is possible [11]. Hence the original benchmarks were implicitly imposing an arbitrary constraint relative to the raw data itself, and the same can apply to imposed graph structure.

As for real-world datasets involving actual multi-table RDB data in its native form, available public benchmarks are somewhat limited and narrow in scope. These include RDBench [79], RelBench [23], and the CTU Prague Relational Learning Repository (CRLR) [54]. However, as of the time of this writing, RelBench constitutes only two datasets, relies on Row2Node, and presents no experiments of any kind; see Appendix H for further differences between RelBench and our work. As for RDBench and CRLR, these are composed mostly of small datasets, e.g., with less than 1000 labeled instances, which is far surpassed by the size of typical real-world RDBs (see Section 4.2 and Appendix H for further details). Additionally, among the recent model-driven works targeting predictive ML or deep learning on RDBs [4, 9, 24, 33, 50, 75, 77], there exists no consistent set of diverse data and tasks for empirical comparisons, and for most there is no available software allowing others to follow suit. See Figure 1(r.h.s.) for a summary of existing benchmark properties.

### 4.2 Benchmark Desiderata and Composition

To increase the chances that strong benchmark performance correlates with strong performance on future real-world application data, it is important to assemble RDB benchmarks so as to achieve adequate diversity or coverage across both (i) datasets and (ii) tasks. With this in mind, on the *dataset side* our selection criteria are as follows: (i) **Availability:** Some otherwise promising public multi-table datasets currently disallow use for research publications [2, 12]; (ii) **Large-scale:** Real-world RDBs can involve billions of rows; (iii) **Domain diversity:** We seek datasets from diverse domains spanning e-commerce, advertising, social networks, etc.; (iv) **Schema diversity:** Variation over schema width, # tables, # of rows; (v) **Temporality**: Realistic RDBs often vary over time.

Meanwhile, on the *task side* we have: (i) **Loss type:** Regression, classification, or ranking; (ii) **Learning type:** Inductive versus transductive; (iii) **Proximity to real-world :** Tasks are chosen to reflect practical business scenarios; (iv) **Meaningful difficulty**: Poorly chosen tasks where informative features are lacking can lead to meaningless comparisons. Conversely, tasks involving auxiliary

features that are simple functions of target labels may be trivially easy. In real-world scenarios, avoiding these extremes may be non-obvious; see Appendix D for representative case studies.

Based on these desiderata covering our proposed 2D dataset and task space underpinning 4DBInfer, we have curated a representative set of RDB benchmarks adhering to Definition 1 in Appendix B. The tasks include: customer retention prediction on **AVS** [18], click-through-rate prediction on Outbrain (**OB**) [53], click-through-rate and purchase prediction on Diginetica (**DN**) [17], conversion prediction on RetailRocket (**RR**) [84], user churn, rating, and purchase prediction on Amazon Book Reviews (**AB**) [55], user churn and post popularity prediction on StackExchange (**SE**) [61], paper venue and citation prediction on **MAG** [59], and customer charge/prepay type prediction on **Seznam** (**SZ**) [54]. For further details, please see Appendices C and D.

Additionally, general comparisons with existing benchmarks are presented in Figure 1, where 4DBInfer displays a distinct advantage in terms of the four overall dimensions we have proposed warrant coverage. As shown in the figure (r.h.s.), relevant existing benchmarks include single-table tabular (**OpenML** [64]), graph (**OGB** [34, 36], **HGB** [51], **TGB** [38]), and RDB (**RDBench** [79], **CRLR** [54], **RelBench** [23]). Note that entity attribute and key prediction correspond with node classification and link prediction in the graph ML literature, respectively. For further reference, Appendix H contains a much broader set of candidate benchmarks that were excluded from 4DBInfer because of failure to adhere with one or more of the above selection criteria.

## 5  Benchmark & Baseline Delivery

To facilitate reproducible empirical comparisons using our proposed benchmarks from Section 4 across the baselines from Section 3 (as well as future/improved predictive models informed by initial results), we instantiate 4DBInfer as a unified, scalable open-sourced Python package; please see `https://github.com/awslabs/multi-table-benchmark/` and Appendix E. This package offers a *no-code* user experience to minimize the effort of experimenting with various baselines over built-in or customized RDB datasets. This is achieved via a composable and modularized design whereby each critical data processing and model training step can be launched independently or combined in arbitrary order. Moreover, adding a new RDB dataset simply requires users to describe its metadata and the location to download the tables; the pipeline will automate the rest. Regarding resource requirements for running current 4DBInfer baselines and benchmarks, including details of peak GPU/CPU memory usage, see Appendix G.1.

As for the critical step of graph sampling, 4DBInfer implements $\Phi$ using the GraphBolt open-source APIs from the Deep Graph Library [68], which facilitates sampling over graphs with billions of nodes, which is roughly tantamount to RDBs with billions rows. Our 4DBInfer toolbox also provides an enhanced implementation of the DFS algorithm to facilitate the large-scale datasets in our benchmark suite. Specifically, the existing open-source implementation FeatureTools [22] can only leverage a single-thread for cross-table aggregation, which can take weeks on some of our datasets. We substitute its execution backend with an SQL-based engine, which translates the feature metadata into SQLs for execution. The resulting solution shortens the DFS computation time to only several hours. Appendix G.2 contains quantitative timing comparisons spanning all of our 4DBInfer baselines and benchmarks.

## 6  Supporting Experiments Using 4DBInfer

We apply our 4DBInfer toolbox to explore performance across the proposed 4D evaluation space defined by benchmarks (datasets and tasks from Section 4) and baselines applied to them (graph extractors/samplers and base predictors from Section 3). As for these baseline models, we explore early feature fusion (DFS-based, with join-path set to reach the farthest RDB tables, i.e., the schema width) and late feature fusion (GNN-based) as discussed in Section 3.3. For the latter, we evaluate the impact of graphs extracted via either Row2Node (**R2N**) or Row2N/E (**R2N/E**) per Section 3.1. Additionally, we also include a widely-used baseline category that involves simply joining information from tables adjacent to the target table in the schema graph, and then applying tabular models to the resulting feature-augmented table. We prefix baselines in this category as **Join** (as a point of reference, this process is analogous to a rudimentary form of 1-hop DFS included within 4DBInfer).

Figure 3 displays a representative summary of performance results applying 4DBInfer, while fine-grained comparisons between every pair of baseline and benchmark are deferred to Appendix A. See also Appendix J for extensive supporting ablations over different graph extraction methods, stronger GNN base predictors, and label usage. While there is considerable detail and nuance associated with these performance numbers, several key points are worth emphasizing as follows:

(a) **Complex vs simple comparisons.** More complex DFS-based and GNN-based models usually outperform simple join base models, indicating that relevant predictive information exists across a wider RDB receptive field (i.e., beyond adjacent tables). These results also highlight the need to consider diverse, relatively large-scale datasets, as prior work [79] involving much smaller scales has shown that simple joins can actually outperform GNNs.

(b) **Early vs late feature fusion.** Early feature fusion as instantiated via GNNs is generally preferable to late fusion through DFS-based models. That being said, DFS nonetheless remains a strong competitor on multiple benchmarks (see Appendix A). Moreover, because of its lean design relative to GNNs, late fusion may be especially favorable in low resource environments even if the accuracy is not necessarily superior.

(c) **Graph extraction method matters.** We observe in Figure 3 that both Row2Node and Row2N/E contribute to high-ranking models, with no clear-cut winner. This is because the preferred graph extraction method depends heavily on the dataset and task (see Appendices A and J.1); hence further exploration along this dimension is warranted.

(d) **Task specific dependencies.** GNNs are preferable for predicting foreign keys (Appendices A and J.2), which is analogous to link prediction tasks in the graph ML literature. The latter typically benefits from more complex structural signals such as common neighbors that GNNs are arguably better equipped to exploit.

In on way or another, each of the points above highlight the value of considering *all four dimensions* of our proposed 4D exploration space, namely, the potential consequences of variability across **dataset (a,b,c)**, **task (d)**, **graph extractor (c)**, and **base predictor (a,b,d)**. Even so, our preliminary comparisons so-obtained crown no unequivocal frontrunner across all scenarios, showcasing the need for such benchmarking on realistic RDB tasks in the first place. And quite plausibly, future high-performant solutions may actually lie at the boundary between tabular and graph ML worlds. Either way, reliably establishing such trends hinges on native RDB evaluations that do not a priori favor one approach over another, e.g., results conditional on only one specific pre-processed graph or feature engineering technique, etc.

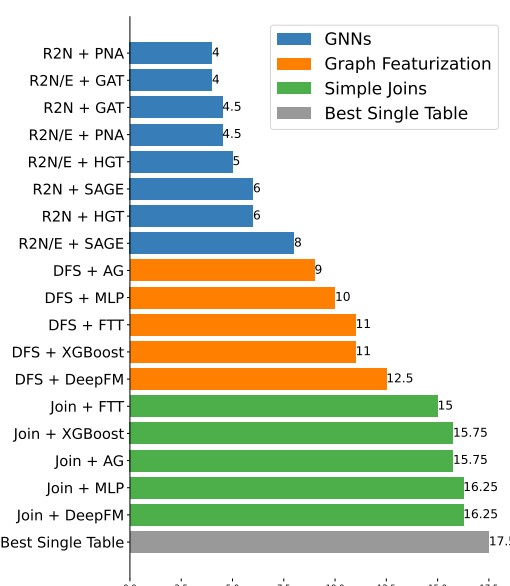

Figure 3: Median performance rank (*lower is better*) of different model combinations (Sec. 3) computed across all 4DBInfer benchmarks (Sec. 4).

## 7 Broader Impact

Large RDBs and/or data lakes composed of multiple high-dimensional tables are pervasive, across which countless opportunities for predictive modeling exist. And yet to do so, data scientists continue to face an unresolved conundrum: Either adopt standard tabular predictive pipelines and then manually supplement with cumbersome cross-table feature engineering, or else rely on deep graph neural network models but struggle with how exactly to form the required input graph from raw tables in the first place. Our 4DBInfer toolbox provides a sturdy bridge between these domains, allowing users to effortlessly train and benchmark scalable solutions across a broad spectrum of possibilities spanning *both* graph and tabular camps.

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

# Appendix Table of Contents

# A   Main 4DBInfer Empirical Comparisons

The results in Table 1 below were aggregated to form Figure 3. Full specification of datasets and tasks are contained in Appendix D, while baseline implementation details can be found in Appendix F. Note that complementary ablation experiments are deferred to Appendix J.

| Dataset | | AVS | OB | DN | | RR | | AB | | SE | | MAG | | SZ | |
|---|---|---|---|---|---|---|---|---|---|---|---|---|---|---|---|
| Task Prediction Type Evaluation metric Induct. or Trans. | | Retent. RA AUC↑ Ind. | CTR RA AUC↑ Ind. | CTR RA AUC↑ Ind. | Purch. FK MRR↑ Ind. | CVR RA AUC↑ Ind. | Churn EA AUC↑ Ind. | Rating RA RMSE↓ Ind. | Purch. FK MRR↑ Ind. | Churn EA AUC↑ Ind. | Popul. EA AUC↑ Ind. | Venue EA Acc.↑ Trans. | Cite FK MRR↑ Trans. | Charge EA Acc.↑ Ind. | Prepay EA Acc.↑ Ind. |
| Single | MLP | 0.5300 | N/A | N/A | N/A | N/A | 0.5000 | N/A | N/A | 0.5000 | 0.5079 | 0.2686 | N/A | 0.4375 | 0.5314 |
| | DeepFM | 0.5217 | N/A | N/A | N/A | N/A | 0.5000 | N/A | N/A | 0.4964 | 0.5078 | N/A | N/A | 0.4242 | 0.5294 |
| | FT-Trans | 0.5013 | N/A | N/A | N/A | N/A | 0.5000 | N/A | N/A | 0.4998 | 0.5124 | 0.2370 | N/A | 0.4367 | 0.5275 |
| | XGB | 0.4505 | N/A | N/A | N/A | N/A | 0.5000 | N/A | N/A | 0.5000 | 0.5036 | 0.2579 | N/A | 0.4289 | 0.5198 |
| | AG | 0.5350 | N/A | N/A | N/A | N/A | 0.5000 | N/A | N/A | 0.5000 | 0.5081 | 0.2547 | N/A | 0.4561 | 0.5145 |
| Join | MLP | 0.5618 | 0.4891 | 0.5082 | 0.0519 | 0.5097 | 0.5000 | 1.0570 | 0.0881 | 0.6024 | 0.8745 | 0.3267 | 0.4989 | 0.5692 | 0.6110 |
| | DeepFM | 0.5620 | 0.5109 | 0.5294 | 0.0502 | 0.4933 | 0.5000 | 1.0585 | 0.0873 | 0.5984 | 0.8764 | 0.2819 | 0.4506 | 0.5416 | 0.5915 |
| | FT-Trans | 0.5569 | 0.5203 | 0.5697 | 0.0612 | 0.4917 | 0.5000 | 1.0574 | 0.0919 | 0.6319 | 0.8670 | 0.2243 | 0.4918 | 0.5825 | 0.6319 |
| | XGB | 0.5545 | 0.5128 | 0.5230 | 0.0526 | 0.5097 | 0.5000 | 1.0550 | 0.0910 | 0.7457 | 0.8678 | 0.2195 | 0.0329 | 0.5581 | 0.6393 |
| | AG | 0.5235 | 0.4965 | 0.5207 | 0.0538 | 0.5000 | 0.5000 | 1.0550 | 0.0853 | 0.7457 | 0.8678 | 0.2571 | 0.3010 | 0.5829 | 0.6290 |
| DFS | MLP | 0.5690 | 0.5456 | 0.7627 | 0.0743 | 0.8181 | 0.6815 | 0.9847 | 0.1112 | 0.8326 | 0.8783 | 0.2887 | 0.4903 | 0.7554 | 0.8248 |
| | DeepFM | 0.5669 | 0.5289 | 0.7260 | 0.0635 | 0.8182 | 0.6667 | 0.9946 | 0.0845 | 0.8212 | 0.8821 | 0.2476 | 0.5760 | 0.7016 | 0.8092 |
| | FT-Trans | 0.5665 | 0.5360 | 0.7430 | 0.0582 | 0.8034 | 0.6765 | 0.9888 | 0.1191 | 0.8376 | 0.8749 | 0.3010 | 0.3635 | 0.7473 | 0.8162 |
| | XGB | 0.5669 | 0.5438 | 0.7612 | 0.0633 | 0.7985 | 0.7371 | 0.9827 | 0.1096 | 0.8308 | 0.8780 | 0.2202 | 0.0329 | 0.7667 | 0.8057 |
| | AG | 0.5646 | 0.5438 | 0.7500 | 0.0749 | 0.8008 | 0.7291 | 0.9829 | 0.0888 | 0.8396 | 0.8849 | 0.3208 | 0.0329 | 0.7731 | 0.8485 |
| R2N | R-GCN | 0.5578 | 0.6239 | 0.6996 | 0.3557 | 0.8470 | 0.7358 | 0.9639 | 0.1790 | 0.8558 | 0.8861 | 0.4336 | 0.7020 | 0.7917 | 0.8768 |
| | R-GAT | 0.5637 | 0.6146 | 0.7570 | 0.3595 | 0.8284 | 0.7410 | 0.9563 | 0.1546 | 0.8645 | 0.8853 | 0.4408 | 0.7072 | 0.8053 | 0.8954 |
| | R-PNA | 0.5606 | 0.6249 | 0.7635 | 0.3638 | 0.8366 | 0.7645 | 0.9615 | 0.1791 | 0.8664 | 0.8896 | 0.5119 | 0.6534 | 0.8000 | 0.8924 |
| | HGT | 0.5703 | 0.6260 | 0.7425 | 0.2207 | 0.8495 | 0.7551 | 0.9636 | 0.1325 | 0.8670 | 0.8817 | 0.4164 | 0.6768 | 0.7965 | 0.8805 |
| R2N/E | R-GCN | 0.5653 | 0.6271 | 0.7191 | 0.3691 | 0.8091 | 0.7207 | 0.9696 | 0.2503 | 0.8485 | 0.6798 | 0.4936 | 0.8065 | 0.7842 | 0.8731 |
| | R-GAT | 0.5638 | 0.6308 | 0.7195 | 0.3746 | 0.7536 | 0.7258 | 0.9657 | 0.3055 | 0.8528 | 0.6883 | 0.5119 | 0.794 | 0.8065 | 0.8963 |
| | R-PNA | 0.5608 | 0.6322 | 0.7223 | 0.3758 | 0.8427 | 0.7348 | 0.9675 | 0.252 | 0.8657 | 0.7045 | 0.5159 | 0.7716 | 0.7988 | 0.8847 |
| | HGT | 0.5630 | 0.6323 | 0.6912 | 0.2072 | 0.8342 | 0.7208 | 0.9663 | 0.2916 | 0.8560 | 0.6603 | 0.4692 | 0.7896 | 0.8071 | 0.8965 |

Table 1: 4DBInfer is informative: Performance results of baselines; the top-5 performing models on each dataset are shaded green - the darker, the better. For abbreviations, EA = Entity Attribute Prediction, RA = Relationship Attribute Prediction, FK = Foreign Key Prediction, Ind. = Inductive, Trans. = Transductive. And some entries are marked as 'N/A' because there are no features in the target table such that single table models cannot be applied.

We remark that although Table 1 does not contain error bars, we have found that trial-to-trial variability is typically quite small for the benchmarks we have chosen. For a representative sample, please see Table 2 with standard errors included across 5 trials. As can be observed from these results, variability is limited to the 3rd or 4th significant digit, which has little impact on performance comparisons.

| Dataset/Task | Metric | DFS/MLP | R2N/RGCN |
|---|---|---|---|
| Amazon/Rating | RMSE↓ | $0.9845 \pm 0.0008$ | $0.9640 \pm 0.0016$ |
| Outbrain/CTR | AUC↑ | $0.5503 \pm 0.0012$ | $0.6239 \pm 0.0026$ |
| Seznam/Charge | Acc↑ | $0.7518 \pm 0.0027$ | $0.7902 \pm 0.0012$ |

Table 2: Results averaged over 5 trials showing modest trial-to-trial variability.

## B  Formal Definition of an RDB Benchmark

**Definition 1.** *We define an RDB benchmark, denoted $\mathcal{B}$, as*

$$\mathcal{B} := \{ \ \{\mathcal{D}_{tr}, \mathcal{I}_{tr}\}, \ \{\mathcal{D}_{val}, \mathcal{I}_{val}\}, \ \{\mathcal{D}_{te}, \mathcal{I}_{te}\}, \ j, \ k \}, \quad where \tag{8}$$

$$\mathcal{D}_{spl} := \{\mathcal{D}(s)\}_{s \in \mathcal{S}_{spl}}, \ \ \mathcal{I}_{spl} := \{\{i, s\} : i \in \psi_{spl}(s), s \in \mathcal{S}_{spl}\}$$

*for all splits labeled $spl \in \{tr, \ val, \ te\}$ that reference training, validation, and testing respectively. For each such split, $\mathcal{D}_{spl}$ includes the database contents at every state $s$ within the set $\mathcal{S}_{spl}$.[3] Meanwhile $\mathcal{I}_{spl}$ contains, for each state $s$ all of the indices $i$ of rows containing the target we wish to predict in column $j$ of table $k$, where $\psi_{spl}(s)$ specifies the set of such indices for each $s$.*

By design, we may readily train baseline models via (3) using $\{\mathcal{D}_{tr}, \mathcal{I}_{tr}\}$ and task specification $\{j, k\}$, while using $\{\mathcal{D}_{val}, \mathcal{I}_{val}\}$ for hyperparameter tuning and model development, reserving $\{\mathcal{D}_{te}, \mathcal{I}_{te}\}$ for final performance evaluations. We also note that Definition 1 accommodates both inductive and transductive learning tasks depending on how $\mathcal{D}(s)$, the sets $\mathcal{S}_{spl}$, and point-to-set mappings $\psi_{spl}$ are defined. Either way, these items are each carefully specified to avoid label leakages, which otherwise represent a significant risk when facing the subtleties of real-world RDBs; see Appendix D.5 for a practical case study that exemplifies how label leakages can unexpectedly occur.

## C  High-Level Datasheets for Datasets

While comprehensive dataset-by-dataset descriptions and deferred to Appendix D, following [26] we provide a high-level overview of their motivation, composition, collection process, preprocessing, uses, distribution, and maintenance.

### Motivation

- **For what purpose was the dataset created?** Section 4.1 explains the motivation behind creating this new benchmark.

- **Who created the dataset and funded the creation of the dataset?** While our 4DBInfer benchmark tasks and preprocessing details are described in Appendix D, the data sources themselves originate as follows:
  - **AVS** and **Outbrain** are Kaggle[4] competitions.
  - **Diginetica** comes from a CIKM 2016 competition [17] sponsored by Diginetica.[5]
  - **RetailRocket** is a Kaggle dataset prepared by Retail Rocket.[6]
  - **Amazon** datasets are prepared by [55].
  - **StackExchange** is prepared using raw data from [61].[7]
  - **MAG** is a dataset prepared by the OGB team [35] from Microsoft Academic Graph [59].
  - **Seznam** is a dataset prepared by CTU Prague [54] from Seznam.[8]

### Composition

---

[3]There may be considerable redundancy across $s$ that can naturally be exploited for efficient storage; however, at least conceptually the notion here is to have access to all relevant $\mathcal{D}(s)$ for each split.

[4]https://www.kaggle.com

[5]https://anyquery.diginetica.com/

[6]https://retailrocket.net/

[7]https://data.stackexchange.com

[8]https://www.seznam.cz/

- **What do the instances that comprise the dataset represent (e.g., documents, photos, people, countries)?** All datasets have multiple instance types, and the specific types are dataset-dependent. Please see Appendix D for details.

- **How many instances are there in total? What data does each instance consist of? Is there a label or target associated with each instance? Are there recommended data splits?** See Appendix D for dataset-specific details.

- **Are relationships between individual instances made explicit?** Yes, all the datasets are relational data where the relationships among different instances are the key consideration.

- **Is the dataset self-contained?** Yes.

- **Does the dataset contain data that might be considered confidential?** No.

- **Does the dataset contain data that, if viewed directly, might be offensive, insulting, threatening, or might otherwise cause anxiety?** No.

- **Does the dataset identify any subpopulations (e.g., by age, gender)?** No.

- **Is it possible to identify individuals (i.e., one or more natural persons), either directly or indirectly (i.e., in combination with other data) from the dataset?** No.

- **Does the dataset contain data that might be considered sensitive in any way?** No.

## Collection Process

- We did not directly collect the original datasets as specified above. Instead, we reprocessed datasets previously published by various vendors. As such, further questions from [26] regarding dataset collection are not applicable in this case.

## Preprocessing/Cleaning/Labeling

- **Was any preprocessing/cleaning/labeling of the data done?** Yes; please see Appendix D for dataset-specific details.

- **Was the "raw" data saved in addition to the preprocessed/cleaned/labeled data (e.g., to support unanticipated future uses)?** No.

- **Is the software that was used to preprocess/clean/label the data available?** No.

## Uses

- **Has the dataset been used for any tasks already?** The original datasets which we repurpose to derive the new benchmark have been used for various purposes, while the specific modifications and task definitions applied to these datasets have not been used anywhere previously.

- **What (other) tasks could the dataset be used for?** The dataset may be used for purposes beyond predictive analysis. For example, one could repurpose the datasets for tabular-based question & answering scenarios.

- **Restrictions on the use of the datasets?** We inherit the restrictions of each original dataset.

## Distribution

- **Will the dataset be distributed to third parties outside of the entity (e.g., company, institution, organization) on behalf of which the dataset was created?** No.

- **How will the dataset be distributed (e.g., tarball on website, API, GitHub)?** The datasets will be accessible via our Python package.

- **When will the dataset be distributed?** The datasets are released along with our toolbox here `https://github.com/awslabs/multi-table-benchmark/`

- **Copyright, license, and IP concerns.** We inherit the licenses of the original data from which we derive our benchmarks.

## Maintenance

- **Who will be supporting/hosting/maintaining the dataset?** The AWS Shanghai AI Lab.

- **How can the owner/curator/manager of the dataset be contacted (e.g., email address)?** min-jiw@amazon.com and quagan@amazon.com
- **Will the dataset be updated?** Yes, we will update them if there are reported issues on GitHub.
- **Will older versions of the dataset continue to be supported/hosted/maintained?** No.
- **If others want to extend/augment/build on/contribute to the dataset, is there a mechanism for them to do so?** They can contact our team on GitHub for discussion.

## D  Comprehensive Dataset and Task Descriptions

| Dataset | # Tables | # Columns | # Rows |
|---|---|---|---|
| AVS | 3 | 24 | 349,967,371 |
| Outbrain (OB) | 8 | 31 | 2,170,441,217 |
| Diginetica (DN) | 5 | 28 | 3,672,396 |
| RetailRocket (RR) | 3 | 11 | 23,033,676 |
| Amazon (AB) | 3 | 15 | 24,291,489 |
| StackExchange (SE) | 7 | 49 | 5,399,818 |
| MAG | 5 | 13 | 21,847,396 |
| Seznam (SZ) | 4 | 14 | 2,681,983 |

Table 3: Statistics of each dataset.

Beyond the high-level datasheets for datasets introduced in Appendix C, this section provides comprehensive details pertaining to each dataset and task originally listed in Table 1. Additionally, for aggregated summary statistics/attributes across each dataset and task, please see Tables 3 and 4, respectively. If not specified, we split training, validation and testing samples according to their timestamps, to simulate real-world scenarios where the trained models will be evaluated over new observed samples. Another design choice is the prediction timestamp, which determines what information in the RDB is available at prediction time. By default, we use the timestamp of the target table as the prediction time, assuming that prediction needs to be made upon a new entry added to the target table. All the datasets have been released as part of the Python package `dbinfer-bench`:

```
pip install dbinfer-bench
```

It can then be loaded from Python:

```python
import dbinfer_bench as dbb
```

### D.1  Acquire Valued Shoppers Challenge (AVS)

```python
dataset = dbb.load_rdb_data('avs')
```

The Acquire Valued Shoppers Challenge [18] is a Kaggle dataset from an e-commerce platform. The dataset has three tables: a `History` table containing the history of each promotion offer given to a customer, an `Offers` table containing the information of the promotion offers themselves, and a `Transactions` table containing the transaction history between customers and products. The schema diagram is shown in Figure 4, where the timestamp column, primary keys, and foreign keys are indicated. Note that the `Customers`, `Chain`, `Category`, `Company` and `Brand` tables are dummy tables that do not exist natively, but are induced from the corresponding foreign key columns [15].

#### D.1.1  Task: Customer Retention Prediction (Retent.)

```python
task = dataset.get_task('repeater')
```

The task given by the dataset vendor is to predict whether a customer will be retained by the platform, i.e. `History.repeater`. Note that in the real world, there are two possible interpretations of the `repeater` column: either a given customer will be retained (hence associated with the customer

| Dataset | Task Description | Prediction Type | Metric | #Train / #Val / #Test | Temporal |
|---|---|---|---|---|---|
| AVS | Customer Retention Prediction (Retent.) | Relationship Attribute | AUC↑ | 109,341 / 24,261 / 26,455 | ✓ |
| Outbrain (OB) | Click-through-rate Prediction (CTR) | Relationship Attribute | AUC↑ | 69,709 / 8,715 / 8,718 | ✓ |
| Diginetica (DN) | Click-through-rate Prediction (CTR) | Relationship Attribute | AUC↑ | 108,570 / 6,262 / 5,058 | |
| | Purchase Prediction (Purch.) | Foreign Key | MRR↑ | 16,247 / 82,721 / 78,357 | ✓ |
| RetailRocket (RR) | Conversion-rate Prediction (CVR) | Relationship Attribute | AUC↑ | 80,008 / 9,995 / 9,997 | ✓ |
| Amazon (AB) | User Churn Prediction (Churn) | Entity Attribute | AUC↑ | 1,045,568 / 149,205 / 152,486 | |
| | Rating Prediction (Rating) | Relationship Attribute | RMSE↓ | 78,485 / 7,762 / 13,492 | |
| | Purchase Prediction (Purch.) | Foreign Key | MRR↑ | 78,485 / 387,914 / 677,211 | ✓ |
| StackExchange (SE) | User Churn Prediction (Churn) | Entity Attribute | AUC↑ | 142,877 / 88,164 / 105,612 | |
| | Post Popularity Prediction (Popul.) | Entity Attribute | AUC↑ | 308,698 / 38,587 / 38,588 | ✓ |
| MAG | Venue Prediction (Venue) | Entity Attribute | Acc.↑ | 629,571 / 64,879 / 41,939 | ✓ |
| | Citation Prediction (Cite) | Foreign Key | MRR↑ | 108,000 / 591,942 / 592,176 | |
| Seznam (SZ) | Charge Type Prediction (Charge) | Entity Attribute | Acc.↑ | 443,276 / 55,410 / 55,410 | |
| | Prepay Type Prediction (Prepay) | Entity Attribute | Acc.↑ | 1,151,620 / 143,952 / 143,953 | ✓ |

Table 4: Benchmark dataset and task details. Note that for foreign key prediction, the validation and test instances include the generated negative samples as well.

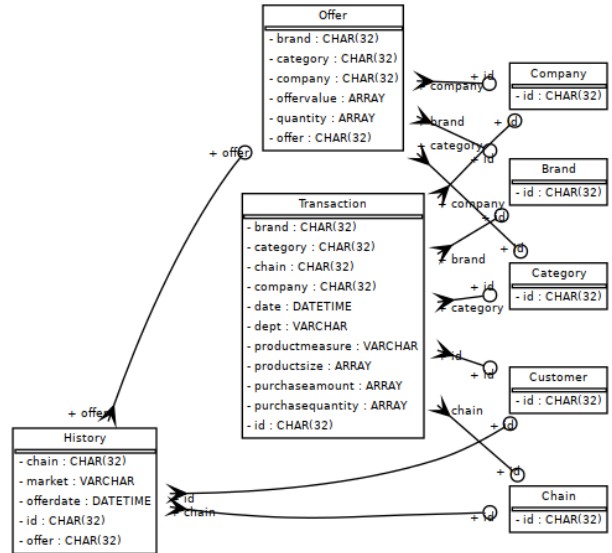

Figure 4: Schema graph for the AVS dataset.

only, making it an attribute of an entity), or else a given customer will repeat the same purchase promoted by the offer (hence associated with the customer and the offer, making it an attribute of a relationship). Since the vendor did not make this distinction clear, we chose the second option. Moreover, since the prediction time is likely different than the date a promotion is offered to a customer, we selected a timestamp later than the offer date.

### D.2 Outbrain Click Prediction (OB)

```
dataset = dbb.load_rdb_data('outbrain')
```

The Outbrain dataset [53] is a large relational dataset from the content discovery platform Outbrain. It contains a sample of users' page views and clicks observed on multiple publisher sites in the United States between June 14, 2016, and June 28, 2016. The dataset consists of several tables. The Events table provides the context information about the user events. The Click table shows which ads were clicked. The Promoted table provides details about the advertise-

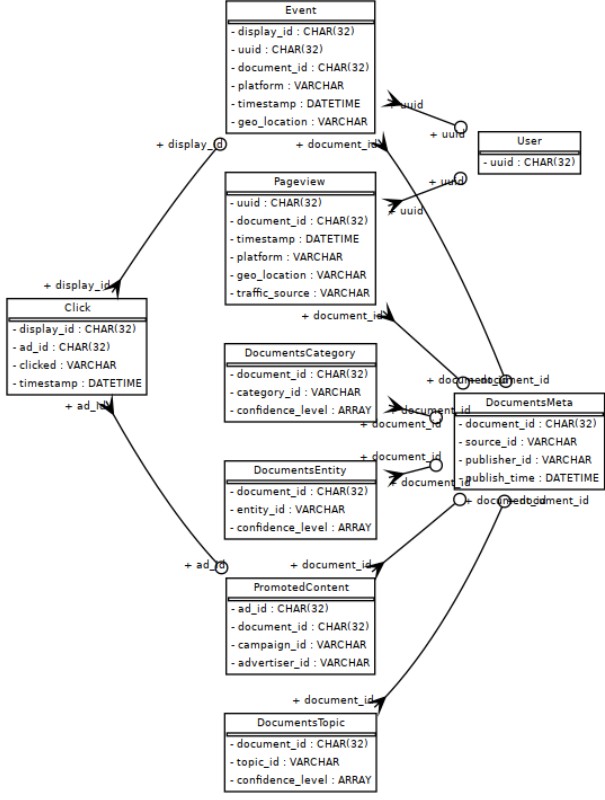

Figure 5: Schema graph for the Outbrain dataset.

ments. The DocumentsCategory, DocumentsTopic and DocumentsEntity provide informa-tion about the promoted contents, as well as Outbrain's confidence in each respective relation-ship. In DocumentsEntity, an entity_id can represent a person, organization, or location. The rows in DocumentsEntity give the confidence that the given entity was referred to in the docu-ment. The dataset schema is shown in Figure 5. Table User is a dummy table induced from the Pageview.uuid and Event.uuid foreign key columns.

### D.2.1 Task: CTR Prediction (CTR)

```
task = dataset.get_task('ctr')
```

The task is to predict whether a promoted content will be clicked or not, i.e. predicting Click.clicked.

### D.3 Diginetica Personalized E-Commerce Search Challenge (DG)

```
dataset = dbb.load_rdb_data('diginetica')
```

The dataset diginetica [17] is part of the Personalized E-commerce Search Challenge and is provided by DIGINETICA and its partners. The dataset focuses on predicting the search relevance of products based on users' personal shopping, search, and browsing preferences. The diginetica dataset consists of several tables: Product contains information about the products, Click contains click data, etc. We conducted the following data cleaning steps from the original tables:

1. The original data does not provide accurate session start times. Instead, the dataset provides the event date and the timeframe (in milliseconds) that each event happens relative to its session. We generate a random start time for each session and convert the relative timeframe of each event into an absolute timestamp.

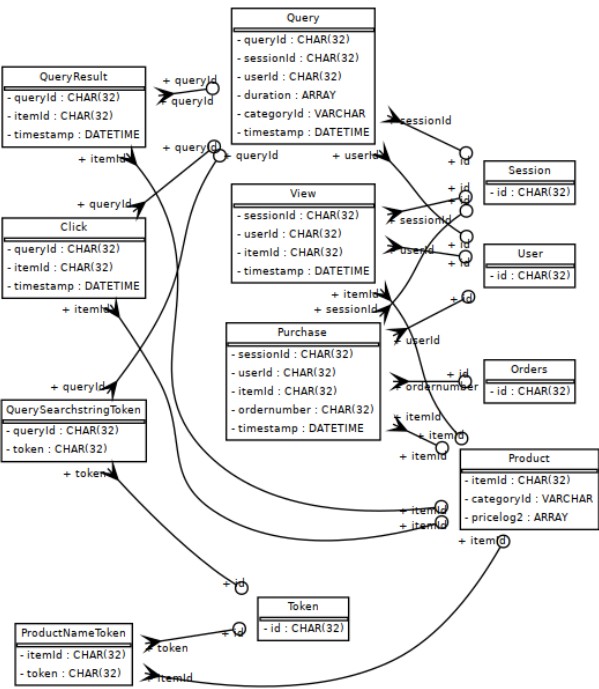

Figure 6: Schema graph for the Diginetica dataset.

2. Product names and query search strings are represented as sequence of anonymous token IDs in the original data. We convert them into two tables `ProductNameToken` and `QuerySearchstringToken`.

3. The original `Query` table stores the query results as a column of item ID lists. We convert that column into a separate table `QueryResult` where each entry is a triplet of `queryId`, `itemId` and `timestamp`.

Steps 2 and 3 make the database satisfy the First Normal Form (1NF) where there are only single-valued attributes [25]. Figure 6 depicts the final RDB schema. Note that `Token`, `Orders`, `Session` and `User` are dummy tables induced from the corresponding foreign keys.

### D.3.1 Task: CTR Prediction (CTR)

```
task = dataset.get_task('ctr')
```

The task is to predict whether an item will be clicked when listed by a given query, i.e., a binary classification task given a triplet of `queryId`, `itemId` and `timestamp`. Positive samples are collected from the `Click` table while negative samples are those in `QueryResult` but not in `Click`. The prediction timestamp is the first time an item is listed by a query to simulate the setting that a recommender system attempts to return the most relevant items for a query. We further down-sample the train/validation/test set to around 100K samples.

### D.3.2 Task: Product Purchase Prediction (Purch.)

```
task = dataset.get_task('purchase')
```

This task is to predict which items will be purchased in a given session, i.e., predicting the foreign key column `Purchase.itemId`. The evaluation metric is Mean Reciprocal Rank (MRR), where the model needs to rank the positive purchase high among 100 randomly generated negative candidates.

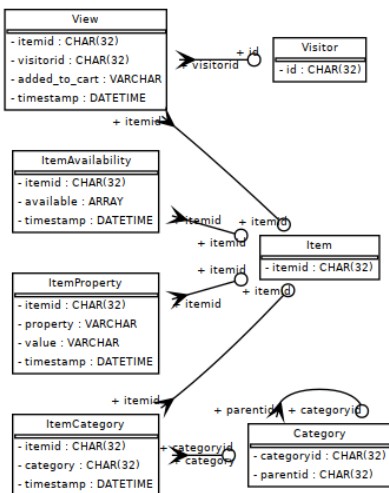

Figure 7: Schema graph for the RetailRocket dataset.

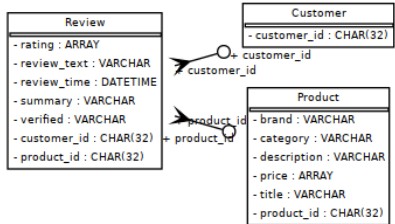

Figure 8: Schema graph for the Amazon Book Reviews dataset.

### D.4 RetailRocket Recommender System Dataset (RR)

```
dataset = dbb.load_rdb_data('retailrocket')
```

The dataset RetailRocket [84] is a Kaggle dataset provided by the E-commerce platform Retail-Rocket. The recorded events represent user interactions on the website. The dataset includes several tables: `View` contains information about whether an item was added to the cart by an user. `Category` stores product category tree. The original dataset stores all item properties in the `ItemProperty` table where most of the property names and values are anonymous tokens. We extract two properties into separate tables: `ItemAvailability` marks the availability status of an item at certain timestamp; `ItemCategory` stores the category information of each item. The dataset schema is shown in Figure 7. Note that `Item`, `Visitor` are dummy tables induced from the corresponding foreign keys.

#### D.4.1 Task: Conversion Rate Prediction (CVR)

```
task = dataset.get_task('cvr')
```

The task is to classify whether an item will be added to the shopping cart by a visitor, i.e. predicting column `View.added_to_cart`. We downsampled the training/validation/testing set to contain 100K samples.

### D.5 Amazon Book Reviews (AB)

```
dataset = dbb.load_rdb_data('amazon')
```

The Amazon Review dataset [55] represents an extensive collection of product reviews on Amazon, encompassing 233 million unique reviews from approximately 20 million users. Our benchmark

utilizes a 5-core subset from the Books category of the original dataset. As depicted in Figure 8, the curated dataset is organized into three tables: The `Customer` table, which catalogues unique IDs for each reviewing customer; the `Product` table, detailing each book with a unique ID, brand, category, description, price, and title; and the `Review` table, documenting each review's connection to a customer and a product, along with the review's rating, text, submission time, summary, and verification status. Spanning from June 25, 1996, to September 28, 2018, this relational database comprises 1.85M customers, 21.9M reviews, and 506K products.

### D.5.1 Task: User Churn Prediction (Churn)

```
task = dataset.get_task('churn')
```

The task is to predict whether a user will continue to engage with the platform and make any purchases in the subsequent three months, forming a binary classification challenge. We select a subset of active users—who have contributed a minimum of 10 reviews in the two years prior to the prediction timestamp—as the set for training, validation and testing.

### D.5.2 Task: Rating Prediction (Rating)

```
task = dataset.get_task('rating')
```

This task is to infer the numerical rating a user might assign to a product, i.e., a regression task on the column `Review.rating`. The prediction should rely solely on the historical review and purchase data, without access to the current review content. The model needs to identify and utilize trends in past user interactions and product engagements to accurately predict the rating, which can range from 1 to 5 stars.

We remark that in practice, when predicting the rating, columns such as `Review.review_text`, `Review.review_time`, `Review.summary` at the same row should not be used, since in real-world settings they are usually given by the customer together with the rating. Hence using these columns to predict a rating at the same row should be treated as a form of information leakage. Nevertheless, it is perfectly fine to use *historical* review texts and summaries to predict the present rating.

### D.5.3 Task: Product Purchase Prediction (Purch.)

```
task = dataset.get_task('purchase')
```

The task is to predict which product will be purchased by a given user, i.e., predicting the foreign key column `Product.product_id`. The evaluation metric is Mean Reciprocal Rank (MRR), where the model needs to rank the positive purchase high among 50 randomly generated negative candidates per positive candidate.

### D.6 StackExchange (SE)

```
dataset = dbb.load_rdb_data('stackexchange')
```

The dataset StackExchange [61] is collected from the online question-and-answer platform StackExchange. The dataset includes several tables: `Badges` includes the information of badges assigned to users; `Comments` stores comments attached to posts; `Posts`, `Tag`, `PostLink` and `PostHistory` are tables of post data; `Users` includes information of users; `Votes` indicates which posts are voted by which users. The dataset schema is shown in Figure 9. Note that prior work [23] has also relied on the same data source for benchmarking. Although we adopt the same task specification, our StackExchange dataset nonetheless remains distinct from [23] in three aspects:

1. We retain the `Tag` table from the raw data source, which contains the linkage to an excerpt post and a Wiki post for each StackExchange tag;

2. We expand the `Tag` attribute in the `posts` table (containing a set of tags for each post) into another table `PostTag`, thereby preserving additional structural information related to post tags;

3. We pull more data from the raw data source, with time-stamps up until 2023-09-03; this augmentation results in 6 months more data than the one used by [23].

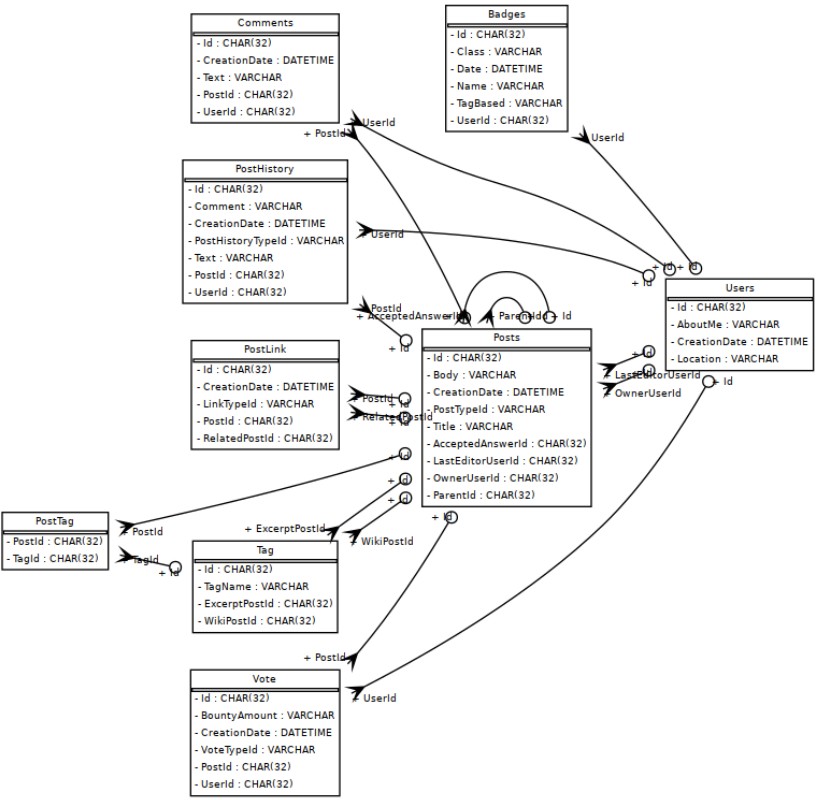

Figure 9: Schema graph for the StackExchange dataset.

### D.6.1 Task: User Churn Prediction (Churn)

```
task = dataset.get_task('churn')
```

The task is to predict whether a user will make any engagement, defined as vote, comment, or post, to the site within the next 2 years starting from year 2011, 2013, 2015, 2017, and 2019 for the training set, and year 2021 for the validation set, and year 2023 for the test set. Note that in the training set, each user requires multiple predictions for different time windows. We also make sure that the prediction timestamps are always later than the user's own creation date.

### D.6.2 Task: Post Popularity Prediction (Popul.)

```
task = dataset.get_task('upvote')
```

The task is to predict whether a given post will be voted in one year since the post was created.

### D.7 Microsoft Academic Graph (MAG)

```
dataset = dbb.load_rdb_data('mag')
```

The dataset is a subset of the Microsoft Academic Graph (MAG), a knowledge graph for academic publications, venues and author information. We include this dataset to highlight the duality between graph and RDB data. We repurposed the `ogbn-mag` dataset from the popular Open Graph Benchmark (OGB) [35] into an RDB of multiple tables: `Paper`, `FieldOfStudy`, `Author` and `Institution` are derived from the four node types; while `Cites`, `HasTopic`, `Writes` and `AffiliatedWith` are created from the four relationships. We retained the same set of node/edge features as table attributes. Specifically, for the `Paper` table, `feat` stores the vector embeddings

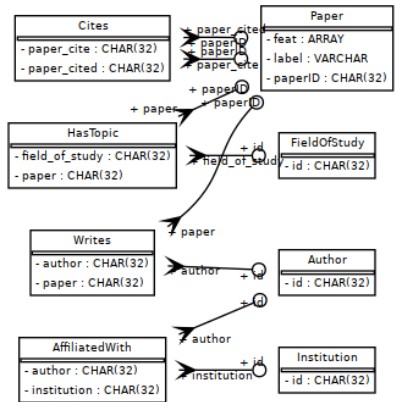

Figure 10: Schema graph for the MAG dataset.

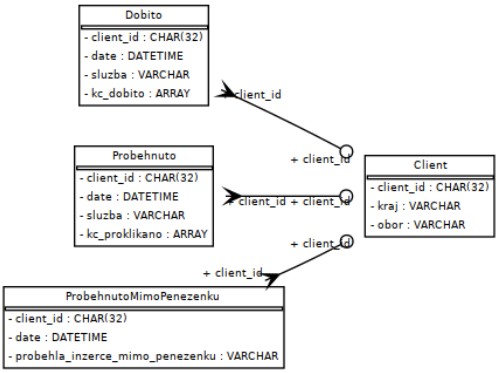

Figure 11: Schema graph for the Seznam dataset.

of each table (processed by OGB) and `label` stores which venue a paper is published at (also the prediction target of Task Venue). The final RDB schema is shown in Figure 10.

### D.7.1 Task: Paper Venue Prediction (Venue)

```
task = dataset.get_task('venue')
```

The task is to predict the venue (conference or journal) a paper is published at. Because the label column is included in the RDB, using the column in making the prediction of the same paper is treated as a form of information leakage, but the model is free to use the labels of connected papers. In total, there are 349 different venues, making the task a 349-class classification problem. Following the practice of `ogbn-mag`, papers are split into train, validation and test sets by their published years.

### D.7.2 Task: Citation Prediction (Cite)

```
task = dataset.get_task('cite')
```

The task is to predict which paper will be cited, i.e., predicting the foreign key column `Cites.paper_cite`. We sampled 100K citations as positive samples, and randomly generated 100 negative candidates for each citation in the validation and test sets. The evaluation metric is Mean Reciprocal Rank (MRR).

### D.8 Seznam (SZ)

```
dataset = dbb.load_rdb_data('seznam')
```

The Seznam dataset [54] is collected from a web portal and search engine in the Czech Republic, which contains online advertisement expenditures from a customer's wallet. The dataset includes the following tables: `Client`, `Dobito` (charges), `Proběhnuto` (prepay), and `ProběhnutoMimoPeněženku` (charges outside the wallet). The `Client` table provides the location and domain field information of the anonymized client. The `Proběhnuto` table includes information about prepayments made into a wallet in Czech currency. The `Dobito` table includes information about charges made from the wallet in Czech currency. The `ProběhnutoMimoPeněženku` table includes information about charges made in Czech currency, but not deducted from the wallet. The RDB schema is shown in Figure 11, where the timestamp column, primary keys, and foreign keys are indicated.

### D.8.1 Task: Charge Type Prediction (Charge)

```
task = dataset.get_task('charge')
```

The task of Charge Type Prediction is to predict the type of each charge, i.e., `Dobito.sluzba`. The evaluation metric for charge prediction is accuracy among 8 classes.

### D.8.2 Task: Prepay Type Prediction (Prepay)

```
task = dataset.get_task('prepay')
```

The task of Prepay Type Prediction is to predict the type of each prepay, i.e., `Proběhnuto.sluzba`. The evaluation metric for prepay prediction is accuracy among 8 classes.

## E    Toolbox Description

We release the benchmarks as a Python package `dbinfer-bench`, installable via PyPI:

```
pip install dbinfer-bench
```

Besides, we also provide a toolbox `dbinfer` for running and comparing the various baselines described in the paper. We modularized the graph-centric predictive pipeline (Figure 2) into several out-of-box command-line tools:

- **Data transform and featurization command** `transform`: Loads an RDB dataset, performs a series of data transformation according to user configurations, and writes the transformed data as a new RDB dataset. The default transformations include: normalizing and canonicalizing various types of feature data (e.g., numerical, categorical, datetime columns, etc.), creating dummy tables, embedding text data into vectors, filtering redundant columns and so on. Users can plug in new data transform logic by inheriting the pre-defined interfaces such as column-wise or table-wise transformer.
- **Deep Feature Synthesis command** `dfs`: Converts an RDB dataset into a new dataset with only a single table, augmented with features produced by the Deep Feature Synthesis algorithm [41]. The command allows users to configure the search depth of the algorithm (setting depth to one gives the simple join baseline), the set of aggregators in use and the backend engine to run (FeatureTools or SQL-based engine).
- **Graph construction command** `construct-graph`: Takes an RDB dataset and produces a graph dataset using algorithms Row2Node (R2N) or Row2N/E (R2N/E).
- **Training tabular-based solution** `fit-tab`: Trains a selected tabular-based solution (e.g., MLP, DeepFM, etc.) over an RDB dataset. Running it over the original RDB dataset corresponds to the single table baseline, while the late-fusion solutions can be launched by running it over the dataset processed by DFS.
- **Training graph-based solution** `fit-gml`: Trains a selected GNN-based solution over a graph dataset.

The toolbox is designed for usability, with a philosophy of omitting unnecessary coding as much of possible. For example, researchers can freely embed some of the commands into their own pipeline, replace some steps with their own, or compose them into new solutions. Moreover, all the commands can be configured via YAML files without modification of the source code. The modularized design also lowers the complexity of fair comparisons among solutions of very different nature. For example, since the tabular-based and GNN-based solutions use the same set of featurization and data transformation steps, it establishes their comparison upon the same foundation. Similarly, researchers can also swap in/out different preprocessing steps to study their impact over the predictive architectures.

# F    Baseline Implementation Details

We conduct standard feature preprocessing for all models such as whitening numeric values, imputing missing entries, embedding text and date/time fields, etc. Where appropriate we also consider including dummy tables; see Appendix J.1. Importantly, to respect the dynamic evolution of RDB datasets, we employ temporal graph sampling, which ensures that only information about preceding events are collected for making predictions (i.e., only information available at RDB state $s$ during which the prediction is being made). All results are collected using the best early-stopping model w.r.t. the validation splits to avoid overfitting. Model-specific details are as follows:

## F.1    DFS-Based Models

Our implementation of DFS leverages Featuretools [22] with the following aggregators: MEAN, MAX, MIN for numeric and embedding features, MODE for categorical features, and COUNT for number of elements per aggregation (i.e. degree). For vector features, we implement custom aggregation primitives as Featuretools does not support it natively. Note that to avoid temporal leakage, for each prediction with timestamp, DFS should not aggregate information from the future beyond the given timestamp (a.k.a. *cutoff time*). Unfortunately, Featuretools does not support cutoff time very efficiently, and obtaining results on some datasets such as RetailRocket is impossible even after 60 hours on an AWS r5.24xlarge instance. While getML's FastProp [27] offers another efficient alternative to Featuretools, the propositionalization engine is implemented in C++ and extending it with custom primitives to support aggregation of embeddings is not trivial. So we developed another solution that translates feature aggregation generated by DFS into SQL, which are then executed by DuckDB.[9]

The predictive models we choose are MLP, DeepFM [31], FT-Transformer [30] and XGBoost [8]. We run XGBoost by invoking TabularPredictor from AutoGluon [21], restricting the candidate models to XGBoost only. For MLP, DeepFM and FT-Transformer, we first project each column into a fixed-length representations using a linear layer, treating categorical variables as one-hot vectors. For MLP, we concatenate all fixed-length representations into a single vector as input. For DeepFM and FT-Transformer, we treat each representation as separate field. The hyperparameter grid is shown in Table 5.

| Hyperparameter | Values |
|---|---|
| Fixed-length representation size | {8,16} |
| Hidden dimension size | {128,256} |
| Dropout | {0, 0.1, 0.3, 0.5} |
| Number of layers | {2, 3, 4} |
| FT-Transformer attention heads | 8 |

Table 5: Hyperparameter grid for DFS-based models.

---

[9]https://duckdb.org/

| Hyperparameter | Values |
|---|---|
| Fixed-length representation size | {8,16} |
| GNN layers | {2,3} |
| Neighbor sampling fanout | {5, 10, 20} |
| Hidden dimension size | {128, 256} |
| Dropout | {0, 0.1, 0.3, 0.5} |
| Number of predictor MLP layers | {2, 3, 4} |
| GAT/HGT attention heads | {4, 8} |
| PNA aggregators | Mean, Min, Max |

Table 6: Hyperparameter grid for GNN-based models.

## F.2 GNN Models

We choose **R-GCN** [57], **R-GAT** [6], **HGT** [37], and **R-PNA** (extending [13] to heterogeneous graphs) as our baselines, with the hyperparameter grid shown in Table 6. The choice of PNA is due to its multitude of aggregators, resembling DFS.

However, the models do not naturally account for edge features. Since Row2N/E could convert some tables to edges and therefore some columns to edge features, we must also extend the aforementioned four models to use edge feature inputs accordingly. The high-level idea is to (1) project the features on each edge $e$ into a fixed-length representation $x_e$, (2) during message passing from a node $u$ to a node $v$ along edge $e$, $x_e$ is also sent along with $u$'s own representation to $v$. The following describes the mathematical details.

To facilitate discussion, denote $u$, $v$ as nodes, and triplet $(u, e, v)$ as an edge connecting from $u$ to $v$ with a unique identifier $e$. Moreover, denote $t(v)$ as the node type of $v$, and $\tau(e)$ as the edge type of $e$.

### F.2.1 R-GCN and R-PNA

R-GCN can be expressed as

$$h_v^{(l)} = \sigma \left( \mathbf{W}_{t(v)}^{(l)} h_v^{(l-1)} + \sum_{(u,e,v) \in \mathcal{N}(v)} c_e \mathbf{W}_{\tau(e)}^{(l)} h_u^{(l-1)} \right), \tag{9}$$

where $\mathcal{N}(v)$ represents the edges going towards $v$ and $c_e$ is some normalization constant associated with edge $e$. To account for edge features $x_e$, we extend it to (changes to the previous equation highlighted in blue)

$$h_v^{(l)} = \sigma \left( \mathbf{W}_{t(v)}^{(l)} h_v^{(l-1)} + \sum_{(u,e,v) \in \mathcal{N}(v)} c_e \mathbf{W}_{\tau(e)}^{(l)} \left[ h_u^{(l-1)} \| x_e \right] \right). \tag{10}$$

We extend R-PNA to handle $x_e$ in a similar fashion.

### F.2.2 R-GAT

R-GAT's formulation is similar to R-GCN's, except that one replaces $c_e$ with a parametrized attention function $\alpha$:

$$h_v^{(l)} = \sigma \left( \mathbf{W}_{t(v)}^{(l)} h_v^{(l-1)} + \sum_{(u,e,v) \in \mathcal{N}(v)} \alpha_{\tau(e)} \left( h_v^{(l-1)}, h_u^{(l-1)}; \mathbf{V}_{\tau(e)}^{(l)} \right) \mathbf{W}_{\tau(e)}^{(l)} h_u^{(l-1)} \right)$$

$$\alpha_{\tau(e)} \left( h_v^{(l-1)}, h_u^{(l-1)}; \mathbf{V}_{\tau(e)}^{(l)} \right) = softmax_u \left[ a_{\tau(e)} \left( h_v^{(l-1)}, h_u^{(l-1)}; \mathbf{V}_{\tau(e)}^{(l)} \right) \right] \tag{11}$$

$$a_{\tau(e)} \left( h_v^{(l-1)}, h_u^{(l-1)}; \mathbf{V}_{\tau(e)}^{(l)} \right) = LeakyReLU \left( \mathbf{V}_{\tau(e)}^{(l)} [h_u^{(l-1)} \| h_v^{(l-1)}] \right).$$

We extend the equations above to

$$
h_v^{(l)} = \sigma \left( \mathbf{W}_{t(v)}^{(l)} h_v^{(l-1)} + \sum_{(u,e,v) \in \mathcal{N}(v)} \alpha_{\tau(e)} \left( h_v^{(l-1)}, h_u^{(l-1)}, x_e; \mathbf{V}_{\tau(e)}^{(l)} \right) \mathbf{W}_{\tau(e)}^{(l)} \left[ h_u^{(l-1)} \| x_e \right] \right)
$$
$$
\alpha_{\tau(e)} \left( h_v^{(l-1)}, h_u^{(l-1)}, x_e; \mathbf{V}_{\tau(e)}^{(l)} \right) = softmax_u \left[ a_{\tau(e)} \left( h_v^{(l-1)}, h_u^{(l-1)}, x_e; \mathbf{V}_{\tau(e)}^{(l)} \right) \right]
$$
$$
a_{\tau(e)} \left( h_v^{(l-1)}, h_u^{(l-1)}, x_e; \mathbf{V}_{\tau(e)}^{(l)} \right) = LeakyReLU \left( \mathbf{V}_{\tau(e)}^{(l)} \left[ h_u^{(l-1)} \| h_v^{(l-1)} \| x_e \right] \right).
$$
$$(12)$$

### F.2.3 HGT

For HGT, we make two changes to make it account for $x_e$: (1) change the *MSG-head$^i$* function in Equation 4 from [37] to additionally concatenate the edge representation $x_e$ before linear projection, (2) in the computation of *ATT-head$^i$* in Equation 3 from [37], add to $W_{\phi(e)}^{ATT}$ a matrix that is a parametrized linear projection of $x_e$.

## G   Toolbox Resource Requirements and Runtime Comparisons

### G.1   Resource Requirements

Most of the baselines provided by the 4DBInfer toolbox can run on an AWS g4dn.8xlarge instance equipped with 128GB CPU RAM, 32 vCPUs, and an NVIDIA T4 GPU with 16GB RAM. Running GNN baselines on the AVS dataset requires 300+ GB CPU RAM to fit the extracted graph in memory. To make testing on the largest dataset OutBrain more affordable, we follow a similar practice as the Illinois Graph Benchmark (IGB) [43], providing a down-sampled version, called OB-Small, which can easily fit within an instance with 128GB CPU RAM. For further reference, Table 7 displays peak GPU and CPU memory usage running 4DBInfer benchmarks using representative baseline methods. For the DFS baseline, we use a 2-hop neighborhood and an MLP prediction head; meanwhile, for the R2N and R2N/E baselines we use an RGCN model.

### G.2   Timing Comparisons

Figure 12 displays how training times roughly grow with model complexity, with DFS-based models taking longer than single-table or simple-join alternatives. Meanwhile, GNNs generally consume the longest training time commensurate with their greater complexity relative to other model families. Notably, we find that GNN models for predicting foreign keys (e.g., DN/Purch., AB/Purch., MAG/Cite) require the longest training; up to 1.5 days, a reasonable duration for many typical research routines. Inference times (not shown) also follow similar trends.

Next, in terms of DFS specifically, Table 8 compares our 4DBInfer pipeline versus the existing FeatureTools implementation [22]. From these results we observe that our SQL-based engine is typically 10x-1000x faster than using FeatureTools.

## H   Further Details Regarding Prior Candidate RDB Benchmarks

Since OpenML [64] contains exclusively single table datasets, and OGB [34, 36], HGB [51], and TGB [38] contain exclusively graphs processed from raw data, we only discuss RelBench, RDBench and CTU Prague Relational Learning Repository in detail.

### H.1   RelBench

RelBench [23] is among a series of contemporary works that highlight the need to transition from graph machine learning to RDBs. As of the time of this writing, RelBench has two datasets: Amazon Book Reviews and StackExchange. Two tasks are designated for Amazon Book Reviews: (1) predicting whether a user will churn beyond a certain timestamp, (2) predicting a user's long term value beyond a certain timestamp. The two tasks designated for StackExchange are: (1) predicting

| Dataset | Task | Baseline | Peak CPU Memory (GB) | Peak GPU Memory (GB) |
|---|---|---|---|---|
| Amazon | Churn | DFS | 23.81 | 0.78 |
| | | R2N | 16.15 | 8.33 |
| | | R2N/E | 27.90 | 14.23 |
| | Purchase | DFS | 23.87 | 1.02 |
| | | R2N | 15.00 | 4.23 |
| | | R2N/E | 27.68 | 11.81 |
| | Rating | DFS | 23.79 | 1.01 |
| | | R2N | 15.27 | 13.00 |
| | | R2N/E | 27.04 | 13.80 |
| AVS | Repeater | DFS | 37.71 | 0.82 |
| | | R2N | 107.50 | 1.03 |
| | | R2N/E | 134.98 | 0.96 |
| Diginetica | CTR | DFS | 5.85 | 0.56 |
| | | R2N | 14.95 | 4.19 |
| | | R2N/E | 12.61 | 1.08 |
| | Purchase | DFS | 5.81 | 0.61 |
| | | R2N | 14.04 | 0.64 |
| | | R2N/E | 12.58 | 0.68 |
| MAG | Cite | DFS | 12.50 | 0.67 |
| | | R2N | 6.49 | 7.61 |
| | | R2N/E | 5.73 | 11.07 |
| | Venue | DFS | 9.93 | 0.80 |
| | | R2N | 5.17 | 4.63 |
| | | R2N/E | 4.24 | 7.48 |
| Outbrain-small | CTR | DFS | 1.37 | 1.01 |
| | | R2N | 2.31 | 3.04 |
| | | R2N/E | 2.08 | 13.74 |
| RetailRocket | CVR | DFS | 9.29 | 0.81 |
| | | R2N | 11.01 | 12.64 |
| | | R2N/E | 11.58 | 2.14 |
| Seznam | Charge | DFS | 1.54 | 0.84 |
| | | R2N | 2.01 | 12.32 |
| | | R2N/E | 2.08 | 3.84 |
| | Prepay | DFS | 1.59 | 0.79 |
| | | R2N | 2.06 | 7.55 |
| | | R2N/E | 2.11 | 12.10 |
| StackExchange | Churn | DFS | 15.87 | 0.93 |
| | | R2N | 5.74 | 13.33 |
| | | R2N/E | 11.77 | 3.04 |
| | Upvote | DFS | 15.93 | 2.30 |
| | | R2N | 4.71 | 4.56 |
| | | R2N/E | 10.45 | 6.11 |

Table 7: Memory usage comparisons across different datasets, tasks, and baselines.

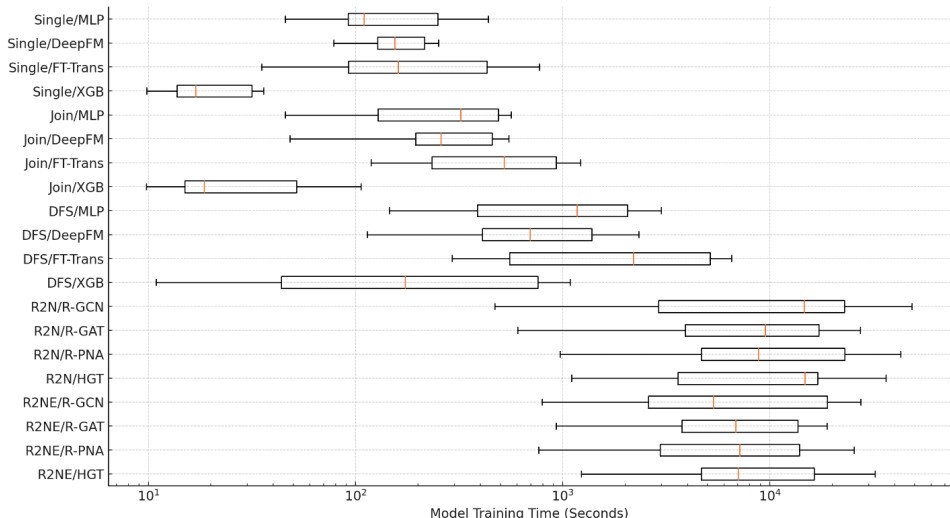

Figure 12: Distribution of model training times across all 4DBInfer benchmarks.

| Dataset / Task | DFS Time (FeatureTools) | DFS Time (ours) |
|---|---|---|
| AVS / Retent. | 647.4 | 175.8 |
| OB (downsampled) / CTR | 16682.43 | 9.78 |
| DN / CTR | > 10 hours | 287 |
| DN / Purch. | > 10 hours | 417 |
| RR / CTR | > 10 hours | 1372 |
| AB / Churn | > 10 hours | 25402 |
| AB / Rating | > 10 hours | 3746 |
| AB / Purch. | > 10 hours | 3773 |
| SE / Churn | > 10 hours | 2802 |
| SE / Popul. | > 10 hours | 2241 |
| MAG / Venue | 6249 | 3812 |
| MAG / Cite | 35 | 99 |
| SZ / Charge | 121.4 | 12.6 |
| SZ / Prepay | 298.2 | 16.2 |

Table 8: Run-time comparisons of 4DBInfer DFS pipeline (sec) versus existing FeatureTools implementation.

whether a user will churn beyond a certain timestamp, (2) predict the number of upvotes a post will receive at a given time window. In all tasks, RelBench designates multiple time windows for every entity to make prediction, effectively making all tasks a time series forecasting task. Presently though, [23] does not come with experimental results.

### H.2 RDBench and CTU Prague Relational Learning Repository

CTU Prague Relational Learning Repository (CRLR) [54] is a collection of 62 real-world and 21 synthetic relational databases. Each relational database comes with one task. However, most of the data are not purposed for a modern RDB machine learning benchmark for the following reasons:

1. Most of the data are dated well before the big data era: only 15 real-world datasets have more than 10,000 labeled instances.

2. The RDB schema is degenerate in that a one-hop outer join can produce a table with no loss of information. An example is the `Airline` dataset.[10]

---

[10]https://web.archive.org/web/20230529190325mp_/https://relational.fit.cvut.cz/dataset/Airline

3. Some tasks are sufficiently easy such that our baselines already achieve 100% accuracy, which compromises our purpose to of using such benchmarks for further advancing RDB machine learning research. The `Accidents` [11] dataset is one such representative example. On the other hand, some tasks have either a very high degree of difficulty or too many noisy or irrelevant features, such that DFS and GNN-based solutions did not show any difference relative to single-table baselines.

4. The business justification of the tasks are not as transparent/direct as the ones typically found in data science competitions. An example is the `IMDB` [12] dataset, where the task is to predict an actor's gender.

Prior work has [79] attempted to address the problems 1 and 2 listed above by handpicking a subset and specifying multiple tasks for each dataset. However, these efforts did not attempt to address problem 3, and some tasks are still arguably too easy, reaching 100% accuracy or 0 regression error.

### H.3 Related Work on Baselines

Although not necessarily framed directly as such, recent work applying predictive ML or deep learning to RDBs can often be interpreted (implicitly or explicitly) as a particular graph extractor ($\mathcal{A}^*$) along with graph-centric sampling ($\Phi$) followed by early [4, 75, 77, 33] or late fusion [9, 41, 48, 47] per the formulation outlined herein.[13] However, there do not as of yet exist systematic comparisons among different pairings of available components (or different graph extraction approaches), nor in most cases is there available code for doing so. In particular, while late fusion-based models (per our terminology) mostly dominate ML solutions on RDBs thus far, more recent GNN-based alternatives (from the early fusion camp) are rarely actually pitted against the strongest incumbents, and vice versa.

As a representative example, the recent RDB benchmarking work from [79] compares GNNs (with graphs from Row2Node) only against tabular baselines involving single tables and 1-hop table joins, not more advanced late fusion approaches like DFS. Conversely, a strong late fusion approach from getML involving more sophisticated joins has recently been compared with GNNs [39], but only against one simple homogeneous GCN architecture [44] that is far from SOTA.

## I Further Details on the Conversion of RDBs to Graphs

A *heterogeneous graph* $\mathcal{G} = \{\mathcal{V}, \mathcal{E}\}$ [63] is defined by sets of node types $V$ and edge types $E$ such that $\mathcal{V} = \bigcup_{v \in V} \mathcal{V}^v$ and $\mathcal{E} = \bigcup_{e \in E} \mathcal{E}^e$, where $\mathcal{V}^v$ references a set of $n^v = |\mathcal{V}^v|$ nodes of type $v$, while $\mathcal{E}^e$ indicates a set of $m^e = |\mathcal{E}^e|$ edges of type $e$. Both nodes and edges can have associated features, denoted $x_i^v$ and $z_j^e$ for node $i$ of type $v$ and edge $j$ of type $e$ respectively. Additionally, if we allow for typed edges linking together arbitrary numbers of nodes possibly greater than two, which defines a so-called hyperedge, then $\mathcal{G}$ generalizes to a heterogeneous *hypergraph* [5, 62], a perspective that will provide useful context below. And finally, for a dynamic graph $\mathcal{G}(s)$, all of the above entities can be generalized to depend on a state variable $s$ as before. We next consider two practical approaches for converting an RDB into a heterogeneous graph (or possibly hypergraph). The motivation here is straightforward: *even if we believe that graphs are a sensible route for pre-processing RDB data, we should not prematurely commit to only one graph extraction procedure*.

### I.1 Row2Node

Perhaps the most natural and intuitive way to convert an RDB $\mathcal{D}$ to a heterogeneous graph $\mathcal{G}$ is to simply treat each row as a node, each table as a node type, and each FK-PK pair as a directed edge. Additionally, non-FK/PK column values are converted to node features assigned to the respective rows. Per this construction, row $\boldsymbol{T}_{i:}^k$ defines a node of type $v = k$. Similarly, if $\boldsymbol{T}_{:j}^k$ represents

---

[11]https://web.archive.org/web/20230128061200/https://relational.fit.cvut.cz/dataset/Accidents

[12]https://web.archive.org/web/20231025130213/https://relational.fit.cvut.cz/dataset/IMDb

[13]There also exist feature augmentation methods based on reinforcement learning that fall outside of our current scope [50, 24]; moreover, scalability and sample-efficiency could pose challenges for such cases.

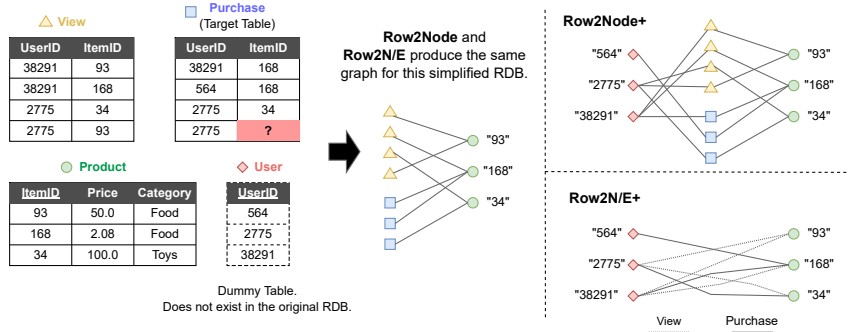

Figure 13: Illustration of Row2Node, Row2N/E and their extended version that includes dummy table.

an FK column of table $k$ that references $\boldsymbol{T}^{k'}_{:j'}$, the PK column $j'$ of table $k'$, then there exists an edge of type $e = kk'$ between the corresponding nodes whenever $\boldsymbol{T}^k_{ij} \rightarrow \boldsymbol{T}^{k'}_{i'j'}$ for row indices $i$ and $i'$. We will refer to this graph composition as *Row2Node* for convenience. As a relatively straightforward procedure for extracting graphs from RDBs, Row2Node was proposed in [15] with ongoing application by others [23, 77, 79]. Figure 13 visualizes the graph constructed by Row2Node from an RDB of three tables: "View", "Purchase" and "Product".

## I.2 Row2N/E

As an alternative to Row2Node, we may relax the restriction that every row must be exclusively converted to a node. Instead, for tables with two FK columns, i.e., $\boldsymbol{T}^k_{:j}$ and $\boldsymbol{T}^k_{:j'}$ are both FKs with $j \neq j'$, we convert each row to an edge connecting the corresponding rows being indexed by the FK pair. More concretely, each such $\{\boldsymbol{T}^k_{ij}, \boldsymbol{T}^k_{ij'}\}$ pair defines an edge of type $e = k'k''$ between rows of tables $\boldsymbol{T}^{k'}$ and $\boldsymbol{T}^{k''}$ as pointed to by $\boldsymbol{T}^k_{:j}$ and $\boldsymbol{T}^k_{:j'}$ respectively. The remaining columns of $\boldsymbol{T}^k$ are designated as edge features. Overall, the intuition here is simply that tables with multiple FKs can be treated as though they were natively a tabular representation of edges.

Additionally, to ensure edges exclusively connect to nodes instead of edges as required in forming a canonical graph, we only convert rows as described above to edges if table $\boldsymbol{T}^k$ has no PK column. If it were to have both two FK columns *and* a PK column, then a referencing FK in yet another table could lead to edge-edge connections which are disallowed by convention.[14] Additionally, we may expand this procedure to generate arbitrary hyperedges [5] by analogously handling tables with three or more FK columns. We henceforth refer to this conversion procedure as *Row2N/E* (short for Row-to-Node-or-Edge), as each row is now selectively treated as either a node or edge/hyperedge depending on the presence of multiple FKs. And if no table within $\mathcal{D}$ has multiple FKs (along with no PK column), then Row2Node and Row2N/E are equivalent. Figure 13 illustrates such a case since both "View" and "Purchases" contain only one foreign key (i.e., "ItemID"). We will see their difference later when more foreign keys are introduced (such as by adding dummy tables).

## I.3 Comparative Analysis

In general there is no ground-truth "correct" graph that can be extracted from an RDB, and hence, no *a priori* gold standard under which we might conclusively prefer Row2Node or Row2N/E, or even something else altogether (e.g., see Section I.4). Nonetheless, there does exist one relevant sanity check with the potential to influence our preferences here. This check relates to a precise form of cycle consistency as follows.

Suppose we are given an initial graph $\mathcal{G}$ as well as a general mapping $\mathcal{A}$ that converts this graph to an RDB via $\mathcal{D} = \mathcal{A}(\mathcal{G})$. We may then apply either Row2Node or Row2N/E to $\mathcal{D}$ and determine

---

[14]In other words, if a table has a PK, then edges from other tables may point to each row; however, if there are also two FK columns, and each row is also converted to an edge, the result would be disallowed edge-edge connections.

if we recover the original $\mathcal{G}$. Ideally, we would like $\mathcal{A}$ to output RDBs in some type of canonical form; otherwise achieving the aforementioned cycle consistency may either be impossible or underdetermined via Row2Node or Row2N/E. As a trivial hypothetical example, the case where $\mathcal{A}$ simply converts every edge of $\mathcal{G}$ to a row within a single table specifying the head node, tail node, relation type, and any associated node/edge features. While the graph is fully specified, it also follows that $K = 1$, there are no FKs pointing to other tables, and both Row2Node and Row2N/E will degenerate to a disconnected graph with a single node type. Hence any meaningful cycle-consistency check must be predicated on a principled choice for $\mathcal{A}$ that precludes such specious possibilities.

Fortunately though, there exist well-established methods for normalizing RDBs into canonical forms [25] that naturally filter out these types of degeneracy and can be repurposed to actualize a reasonable idempotency check.

**Proposition 1.** *Let $\mathcal{G}$ denote a heterogeneous graph and $\mathcal{A}$ a mapping that converts $\mathcal{G}$ to a degenerate single table RDB as described above. Furthermore, let Norm denote an operator that normalizes an RDB according to the first through forth database normal forms.[15] Then Row2Node and Row2N/E as specified in Sections I.1 and I.2 are such that*

$$
\begin{aligned}
\mathcal{G} &\neq Row2Node\,[\,Norm\,(\,\mathcal{A}[\mathcal{G}]\,)\,] \\
\mathcal{G} &= Row2N/E\,[\,Norm\,(\,\mathcal{A}[\mathcal{G}]\,)\,].
\end{aligned}
\tag{13}
$$

Informally, Proposition 1 demonstrates that Row2N/E has an advantage in terms of recovering a ground-truth graph that has been converted to a properly normalized/standardized RDB as quantified by well-studied database normal forms. Of course we cannot necessarily infer from this that Row2N/E is broadly preferable. Even so, this result is one noteworthy attribute worthy of consideration. Beyond this, we remark that Row2Node and Row2N/E can also be related through the notion of star-graph expansions of hypergraphs [82] via

$$
\begin{aligned}
&Row2Node\,[\,Norm\,(\,\mathcal{A}[\mathcal{G}]\,)\,] \\
&\quad = Star\,(\,Row2N/E\,[\,Norm\,(\,\mathcal{A}[\mathcal{G}]\,)\,]\,).
\end{aligned}
\tag{14}
$$

In this expression $Star(\cdot)$ produces the star-graph expansion of an arbitrary input hypergraph $\mathcal{G}$. Although star-graph expansions have limitations [72], they are nonetheless widely used to process hypergraphs [1, 71]. That being said, as will be discussed in Section 3.3 and empirically tested in Section 6, predictive baseline models built upon Row2Node and Row2N/E need not perform the same even under the restrictive setting of input RDBs constructed as $\mathcal{D} = Norm\,(\,\mathcal{A}[\mathcal{G}]\,)$; likewise for more diverse regimes/RDBs where generally $Row2Node\,[\mathcal{D}] \neq Star\,(\,Row2N/E\,[\mathcal{D}]\,)$.

## I.4   Extension to Row2Node+ and Row2N/E+

Thus far we have assumed that when converting an RDB to a graph, edges are exclusively formed by *known* PK-FK pairs. But practical use cases (as reflected in the benchmarks we will introduce later) sometimes warrant the invocation of another less obvious type of edge formation. Specifically, although not explicitly labeled as such, within an RDB there may exist one or more table columns with the signature of an FK (e.g., elements are a high cardinality categorical index or related), but with *no associated PK column in another table*. Moreover, the RDB may also contain a second column with elements drawn from the same high-cardinality domain; this additional column may reside in either the same or a different table as the original. Together these *pseudo FK* columns can be converted to actual FKs by introducing a new dummy table, with just a single column treated as a PK, defined by the unique corresponding elements of the pseudo FKs. In this way extracted graphs have additional pathways for sharing information across or within the original tables by passing through nodes associated with the new dummy table.

Note that this conversion of pseudo FK pairs (or the natural extension to arbitrary FK tuples) can be integrated within either Row2Node or Row2N/E, and we henceforth refer to these variants as *Row2Node+* and *Row2N/E+* respectively. In the example depicted in Figure 13, the column "UserID" is a pseudo FK, which if converted to an actual FK, gives birth to an additional *dummy* "User" table. Consequently, Row2N/E+ further treats entries in the "View" and "Purchase" table

---

[15]We remark that the first four normal forms are the most common normalizations used in practice [52]. For further details on database normalization and normal form definitions, we refer the reader to [25].

as edges, resulting in different graph constructed than Row2Node+. Please see Appendix J.1 for ablations using both Row2Node+ and Row2N/E+.

In practice there is no strict objective standard for when to convert pseudo FK tuples into dummy tables and new FK-PK pairs as described above; however, the process of finding candidates for such a conversion closely mirrors the notion of Joinable Table Discovery (JTD) [19]. Although JTD is often applied on data lakes with a vast number of tables, the same logic can also be applied on RDBs to discover such pseudo FK tuples, if one treats RDB as a "small" data lake.

### I.5  Proof of Proposition 1

Given a heterogeneous attributed graph $\mathcal{G}$, we can always construct a single table $\bar{T} = \mathcal{A}(\mathcal{G})$, via an injective mapping $\mathcal{A}$, such that the $i$-th row satisfies

$$\bar{T}_{i:} = [u, w, v_u, v_w, x_u, x_w, e_i, z_i], \tag{15}$$

where $u$ and $w$ represent the head and tail node indeces of edge $i \in \mathcal{E}$. Moreover, with some abuse of notation $\{v_u, v_w\}$ and $\{x_u, x_w\}$ represent the corresponding node types and node features, respectively. Meanwhile, $e_i$ and $z_i$ indicate the relation type and (optional) feature of edge $i$. Additionally, as $\mathcal{G}$ is a heterogeneous graph (as opposed to multi-graph), each triplet $\{u, w, e_i\}$ of head node, tail node, and relation type is unique and serves as a candidate key for the table. (Note that node and edge features, as well as node types, cannot contribute to a candidate key as we make no assumptions on their values, e.g., they could all in principle be equal or non-distinguishing.)

From here, by assumption $\bar{T}$ will have unique rows such that it satisfies the criteria for an unnormalized form (UNF), i.e., no duplicated rows. Next, provided we treat each node and edge feature as a single entity, then the first normal form (1NF) is satisfied (if we were to treat each feature as a set or nested record, technically it would not, but for our purposes this distinction is inconsequential). Proceeding further, to address the second normal form (2NF) we examine all non-candidate key attributes to determine which are dependent on the entire candidate key and which are not. Clearly $z_i$ does in fact depend on the entire candidate key, and so it satisfies 2NF. Notably though, $x_u$ and $v_u$ only depend on $u$, while $x_w$ and $v_w$ only depend on $w$; in neither case is there dependency on the entire candidate key. Hence to satisfy 2NF, we must form a second table to record non-duplicated records of node features and node types, and remove these attributes from $\bar{T}$. Hence rows of $\bar{T}$ simplify to

$$\bar{T}_{i:} = [u, w, e_i, z_i] \tag{16}$$

and we introduce the node attribute table $\bar{T}^{node}$ with row $u$ given by

$$\bar{T}^{node}_{u:} = [u, v_u, x_u]. \tag{17}$$

Both $\bar{T}^{node}$ and $\bar{T}$ now satisfy 2NF, with $u$ serving as the primary key for the former, while $u$ and $w$ now independently serve as foreign keys for the latter. Next, as there are no transitive functional dependencies, nor multivalued dependencies, the third normal form 3NF and forth normal form (4NF) are trivially satisfied. We may therefore conclude that our new tables satisfy

$$\{\bar{T}, \bar{T}^{node}\} = \text{Norm} \left( \mathcal{A}[\mathcal{G}] \right) \tag{18}$$

per our previous definitions.

From here we observe that Row2Node will introduce new nodes associated with each row of *both* $\bar{T}$ and $\bar{T}^{node}$, with the former *not* present in the original $\mathcal{G}$. From this it follows that

$$\mathcal{G} \neq \text{Row2Node} \left[ \text{Norm} \left( \mathcal{A}[\mathcal{G}] \right) \right]. \tag{19}$$

As for Row2N/E, because of the newly-introduced FK-PK relationship, it naturally follows that

$$\mathcal{G} = \text{Row2N/E} \left[ \text{Norm} \left( \mathcal{A}[\mathcal{G}] \right) \right], \tag{20}$$

completing the proof.

## J   Additional 4DBInfer Ablation Experiments

The ablations included within this section can be summarized as follows:

- **Broadening graph extraction using dummy tables.** In Appendix I, and in particular I.4, we described how the strategic introduction of so-called dummy tables can lead to extracted graphs with additional inter- or intra-table edges. Appendix J.1 below compares across identical settings with and without the use of such dummy tables; in many cases there is a significant performance impact, e.g., for R-GCN models using Row2Node on AVS-Retent the AUC drops from 0.5653 to 0.4761 without dummy tables (a similar drop also occurs when using Row2N/E).

- **Stronger GNN model.** We examine the extent to which more recent GNN architectures might further boost performance. For this purpose, we conduct experiments using neural common neighbors (NCN) [69], a powerful architecture specifically targeting link prediction. As detailed in Section J.2, on 7 of 8 benchmarks related to key or relationship attribute prediction, NCN improves upon all of the baselines in Table 1.

- **Label propagation.** Finally, as alluded to in Section 2.1, it is possible to handle trainable generalizations of label propagation using the conceptual framework that underpins 4DBInfer. We explore this possibility in Appendix J.3, demonstrating that the judicious use of observable labels can positively influence performance by significant margins, e.g., without such use of labels the AUC can drop by over 0.10 on the OB-CTR task.

## J.1 Graph Extraction With and Without Dummy Tables

In accordance with the descriptions of each dataset and task (Appendix D), we introduce dummy tables with PK columns matched with high-cardinality columns in original RDB tables (an example is the "Customer" table in the AVS dataset). More concrete details were discussed in Appendix I.4, where Row2Node+ and Row2N/E+ were introduced as two alternative graph extraction approaches (built on Row2Node and Row2N/E, respectively) that incorporate dummy tables. Introducing dummy tables can enrich inter-row connections, with multiple implications. Firstly, adding dummy tables establishes new FK-PK relationships, from which DFS may obtain more features. Secondly, dummy tables introduce extra node types and edge types that can be leveraged by GNN models.

To study its impact on model performance, we choose five datasets involving dummy tables, and evaluate the performance changes for both DFS solutions and GNN solutions, with and without the presence of dummy tables. In order to remove the dummy tables, for RR and MAG, we directly drop the FK columns referring to the dummy tables. We choose this approach instead of converting these columns into categorical features in order to maintain an inductive setting. However, for the AB and AVS datasets, directly dropping these columns would result in changes to task specification. Hence, in these cases, we treat them as categorical features and drop the dummy tables.

Table 9 shows the results with and without dummy tables. For DFS with MLP models, we only report the results on AB and AVS. This is because, for the other datasets, the generated features remain unchanged regardless of the presence of dummy tables. It is evident that the performance significantly decreases without the inclusion of dummy tables, underscoring the importance of creating these tables. The situation is more complex for GNN solutions. Removing the dummy tables consistently leads to a decrease in performance for the AB, AVS, and MAG datasets. However, for results on OB datasets, there appears to be no significant difference in performance with or without dummy tables. One possible explanation is that the related columns have less importance, resulting in minimal benefits from the addition of dummy tables. Nevertheless, for certain datasets, creating dummy tables can enrich the connection relationships and effectively enhance performance.

## J.2 Stronger Task-Specific GNN - Neural Common Neighbors

To explore the possibility of stronger GNN architectures on a task-specific basis, we have implemented a powerful link prediction method based on the Neuron Common Neighbor (NCN) algorithm [69]. NCN is an effective approach for link prediction that includes the incorporation of common neighbor embeddings into the prediction process. This method has distinct advantages in terms of expressiveness, as common neighbors cannot be expressed in traditional message-passing GNN, and it has previously been shown to achieve state-of-the-art performance. We have conducted tests using NCN for all 8 relation attribute and foreign key tasks, and the results indicate that NCN performs quite well. Furthermore, the results demonstrate that RDB benchmark graphs still maintain certain

| Dataset/Task | | AB/Rating | AVS/Retent. | OB/CTR | MAG/Venue |
|---|---|---|---|---|---|
| Prediction Type | | RA | RA | RA | EA |
| Evaluation Metric | | RMSE↓ | AUC↑ | AUC↑ | Acc.↑ |
| DFS MLP | w/ dummy | 0.9847 | 0.5690 | - | - |
| | w/o dummy | 1.0291 | 0.5469 | - | - |
| R2N R-GCN | w/ dummy | 0.9639 | 0.5653 | 0.6239 | 0.3792 |
| | w/o dummy | 1.0495 | 0.4761 | 0.6173 | 0.3762 |
| R2N R-GAT | w/ dummy | 0.9563 | 0.5637 | 0.6146 | 0.3888 |
| | w/o dummy | 1.0493 | 0.4687 | 0.6160 | 0.3771 |
| R2N/E R-GCN | w/ dummy | 0.9696 | 0.5653 | 0.6271 | 0.4671 |
| | w/o dummy | 1.0536 | 0.4708 | 0.6244 | 0.4111 |
| R2N/E R-GAT | w/ dummy | 0.9657 | 0.5638 | 0.6308 | 0.4512 |
| | w/o dummy | 1.0511 | 0.4708 | 0.6328 | 0.4071 |

Table 9: DFS and GNN performance comparisons with or without dummy tables. Note that Row2Node+ and Row2N/E+, respectively, are used in Appendix I.4 to reference Row2Node and Row2N/E graph extraction methods augmented with dummy tables.

characteristics when compared to general graph tasks, suggesting that further research on GNN can be applied to the RDB benchmark.

To restate the link prediction problem: link prediction is a widely studied issue in graph analysis, whereby a model is tasked with predicting the presence or absence of a link between two given nodes. In the context of the RDB benchmark, this problem encompasses two tasks: relation attribute prediction and foreign key prediction.

Since the original NCN method exclusively applies to static homogeneous graphs, we have expanded it to encompass temporal and heterogeneous settings. The primary process can be divided into three supplementary steps compared to simple GNN baselines: specifying common neighbor meta-paths, retrieving common neighbor embeddings in GNN layers, and appending common neighbor embeddings. Suppose that we are given two nodes $i_0$ and $j_0$, having node types $v_{i,0}$ and $v_{j,0}$ respectively. Common neighbor meta-paths consist of two sequences of node types $[v_{i,0}, v_{i,1}, \ldots, v_{i,k_i}]$ and $[v_{j,0}, v_{j,1}, \ldots, v_{j,k_j}]$, where $v_{i,k_i} = v_{j,k_j}$ denotes the common neighbor node type, and $k_i$ and $k_j$ represent the number of common neighbor hops from $i$ and $j$. The two meta-paths specify how we discover common neighbors. We sample neighboring nodes of $i_0$ and $j_0$ with node types based on their corresponding meta-paths until we reach the final hop. Subsequently, we assess whether the sampled nodes are the same, implying that they are common neighbor nodes. Formally, we search for nodes $Common\_neighbors = \{i_{k_i} = j_{k_j}\}$ such that there exist nodes $[i_1, i_2, \ldots, i_{k_i}]$ satisfying $i_t \in \mathcal{N}_{i_{t-1}}^{v_{i,t-1}, v_{i,t}}, t \in [1, \ldots, k_i]$, and $[j_1, j_2, \ldots, j_{k_j}]$ satisfying $j_t \in \mathcal{N}_{j_{t-1}}^{v_{j,t-1}, v_{j,t}}, t \in [1, \ldots, k_j]$. Once we have identified the common neighbors, our goal is to retrieve their embeddings. One simple approach is to re-run GNN on the common neighbor nodes. However, we adopt a more efficient method by directly utilizing the inner GNN embeddings as the common neighbor embeddings. It is important to note that the common neighbor nodes are also sampled during the message passing, and their intermediate embeddings are also computed. Therefore, we can utilize the corresponding inner GNN embeddings $\boldsymbol{h}_{i_{k_i}, \ell-k_i}^{v_{i,k_i}}$ and $\boldsymbol{h}_{j_{k_j}, \ell-k_j}^{v_{j,k_j}}$. The final embedding for prediction is derived from

$$MLP(\boldsymbol{h}_{i,\ell}^{v_{i,0}} || \boldsymbol{h}_{j,\ell}^{v_{j,0}} || \sum_{i_{k_i} \in Common\_neighbors} \boldsymbol{h}_{i_{k_i}, \ell-k_i}^{v_{i,k_i}} || \sum_{j_{k_j} \in Common\_neighbors} \boldsymbol{h}_{j_{k_j}, \ell-k_j}^{v_{j,k_j}}).$$

We have implemented NCN based on R-GCN and conducted tests on it for some Relation Attribute and Foreign Key tasks. The results are presented in Table 10 (with the best performance in boldface), where NCN outperforms R-GCN, suggesting the potential for stronger link prediction models. An intriguing phenomenon arises as the performance of NCN varies depending on the graph construction method employed, but generally, it exhibits superior performance for Row2N/E. One plausible explanation is that Row2N/E generates a denser graph, thus making it easier to discover common neighbors. Performance differences due to different graph structures are also worthwhile as a future work.

| Dataset | | AVS | OB | RR | AB | | MAG |
|---|---|---|---|---|---|---|---|
| Task | | Retent. | CTR | CTR | Rating | Purch. | Cite |
| Prediction Type | | RA | RA | RA | RA | FK | FK |
| Evaluation metric | | AUC↑ | AUC↑ | AUC↑ | RMSE↓ | MRR↑ | MRR↑ |
| R2N | R-GCN | 0.5578 | 0.6239 | 0.8470 | 0.9639 | 0.1790 | 0.6336 |
| | NCN | 0.5613 | 0.6189 | **0.8620** | 0.9750 | 0.1944 | 0.6634 |
| R2N/E | R-GCN | 0.5653 | 0.6271 | 0.8091 | 0.9696 | 0.2503 | 0.7539 |
| | NCN | **0.5658** | **0.6294** | 0.8156 | **0.9638** | **0.3071** | **0.8147** |

Table 10: Results of NCN

## J.3 Propagating Labels

In the context of classification and regression tasks, the incorporation of label information from related instances can serve as a strong signal and indicator for predicting the target. We add label information to our pipeline and consider it as a normal feature, performing label propagation in the message passing process to enhance the predictive power. With the help of the temporal neighbor sampler in our framework, any future information will be filtered out, which avoids label leakage when introducing label information.

We now examine the effectiveness of propagating labels from two angles. We select four tasks and assess the changes in performance of R-GCN under different settings. Results are presented in Table 11. Firstly, we remove the label information and only rely on features for prediction. Compared to the default setting which unitizes both label and features, this leads to a significant decrease in performance, underscoring the importance of incorporating label information. Secondly, we remove all features and retain only the label information and data structure, establishing a pure label propagation setting. As a result, in three out of four tasks, there is no noticeable decline in performance. For the RR-CTR task, although the AUC decreases by approximately 3%, the performance of only utilizing labels is still superior to that of only using features.

Based on the comparison of these three sets of results, we can conclude that related labels are often quite informative and can be a very discriminative feature. Propagating labels can therefore serve as an effective and essential approach for enhancing predictions in many practical scenarios.

| Dataset/Task | | AB/Rating | RR/CTR | OB/CTR | MAG/Venue |
|---|---|---|---|---|---|
| Prediction Type | | RA | RA | RA | EA |
| Evaluation Metric | | RMSE↓ | AUC↑ | AUC↑ | Acc.↑ |
| R2N | Default | 0.9639 | 0.847 | 0.6239 | 0.4338 |
| | Remove label | 1.0064 | 0.7808 | 0.5233 | 0.3921 |
| | Remove features | 0.9742 | 0.817 | 0.6238 | 0.4329 |
| R2N/E | Default | 0.9696 | 0.8091 | 0.6271 | 0.4895 |
| | Remove label | 1.0057 | 0.7554 | 0.4952 | 0.4615 |
| | Remove features | 0.9566 | 0.7818 | 0.6227 | 0.5033 |

Table 11: Results of propagating labels.

