# OpenReview forum: "4DBInfer:  A 4D Benchmarking Toolbox for Graph-Centric Predictive Modeling on RDBs"
_NeurIPS.cc/2024/Datasets_and_Benchmarks_Track — NeurIPS 2024 Track Datasets and Benchmarks Poster_

### Official Review · Reviewer_RpkB · 2024-06-27
**Review of submission 705**

**Rating:** 7
**Confidence:** 3

**Review:**

The paper introduces a new RDB benchmark that includes multiple DBs and relevant prediction tasks; it also introduces a framework to apply predictive modeling on RDBs using GNNs.

An extensive experimental campaign highlights the effectiveness of GNN-based methods on the various tasks when compared with traditional, join-based methods. The benchmark is publicly available and it includes extensive documentation and examples. Importantly, providing sizeable databases and their associated tasks should help with benchmarking general use RDB and help foster research on this subject.

The main weaknesses of this work lie mainly in its presentation. It is a very dense paper, featuring very wordy (and sometimes nonsensical) writing. It is also clearly biased towards GNNs, ignoring both some of their known drawbacks, and non-GNN alternatives. Finally, there is no detail on the computational resources required to run the benchmark.

I believe this is solid work, however it could be further improved by addressing the points raised above.

**Strengths:**

- The benchmark is clearly a valuable contribution as it features open, very large RDBs with clear associated tasks. This will likely be very significant in improving the quality of future research on RDB methods.
- The experimental section is extensive and includes interesting baselines and ablation.
- The documentation and appendix provide a lot of insight into how to run the code and what to expect.

**Additional Feedback:**

I was unable to install the package in the repository on my machine using pip because of the DGL requirement being impossible to satisfy.

**Clarity:**

The paper is not well written. In fact, it is very wordy and heavy to follow.

Various sentences are overly convoluted (e.g. "At a high-level, once granted \Phi we sub-divide candidate architectures for instantiating the predictive distribution from (3) based on what can be loosely referred to as early versus late feature fusion."), or simply do make sense ("Our 4DBInfer toolbox for pairing a so-called 2D design space of baseline models from Section 3 and the 2D benchmark coverage from Section 4 within a neutral combined 4D evaluation setting is introduced in Section 5.").

I strongly encourage the authors to lighten the text; doing so would have the added benefit of freeing up space that could be used to move some of the content of the appendix to the main body.

**Correctness:**

The benchmarking RDBs have been prepared in a convincing way.

The experimental section is exhaustive and includes ablation and extensive explanations of the experiments. However, it lacks information on the computational resources.

**Documentation:**

The repository provided by the authors includes exhaustive documentation, which is complemented by the additional details provided in the appendix to the paper.

**Ethics:**

No ethical concerns.

**Limitations:**

The authors do not address the downsides of GNNs, nor do they provide adequate information on the computational resources required to run experiments with the benchmark. Providing info on the compute required would help gauge the different methods better, and inform practitioners of the kind of resources that should be allocated to run the benchmark.

Evaluation of prior work on graph embeddings is lacking and leads to a biased view of the field. Discussing how prior work could not be applied to the problem at hand would strengthen the submission and the argument for using GNNs.

**Opportunities For Improvement:**

While the current evaluation is low, it will be raised appropriately if the following points will be suitably addressed.

W1. The paper include an extensive section on prior work on GNNs. However, it does not include any discussion on alternative methods for generating graph embeddings ([1], [2], [3]), or prior work on converting tabular data into graph form ([4], [5]).
This gives the impression of a very obvious bias towards GNNs, and it is especially jarring to see the statement "Even if we believe that graphs are a sensible route for pre-processing RDB data, we should not prematurely commit to only one graph extraction procedure and the coincident downstream inductive biases that will inevitably be introduced." If possible, I would strongly encourage the authors to try evaluate some of the simpler suggested baselines to make the submission more convincing. A strong comparison against baselines would also make the case for the submission stronger.

W2. There is no mention of the computational resources that were used to conduct the experiments (CPU/GPU/RAM usage). As is, it is not clear whether the benchmark must be installed on a server with extensive computational resources, or if it could run on a laptop. Please add a section on the resources required to the main body of the paper.

W3. The paper is not well written, which makes it hard to follow and confusing at times.
- The abstract begins by mentioning the prediction of missing column values in a target table (an imputation task), but from the body it is not clear whether this is the task that is being evaluated in the benchmark, or the objective used to train the GNN. The tasks included in the benchmark do not seem to involve the prediction of missing values either. Please clarify this point.
- In Figure 1, what does "billion-scale" mean? Is it the number of rows, the number of cells, the size in KB? Please make it more explicit.

Refs.
[1] Grover, A., & Leskovec, J. (2016, August). node2vec: Scalable feature learning for networks. In _Proceedings of the 22nd ACM SIGKDD international conference on Knowledge discovery and data mining_ (pp. 855-864).

[2] |Chen, Haochen, et al. "Harp: Hierarchical representation learning for networks." _Proceedings of the AAAI conference on artificial intelligence_. Vol. 32. No. 1. 2018.|
|APA||

[3] Perozzi, B., Al-Rfou, R., & Skiena, S. (2014, August). Deepwalk: Online learning of social representations. In _Proceedings of the 20th ACM SIGKDD international conference on Knowledge discovery and data mining_ (pp. 701-710).

[4] Cappuzzo, R., Papotti, P., & Thirumuruganathan, S. (2020, June). Creating embeddings of heterogeneous relational datasets for data integration tasks. In _Proceedings of the 2020 ACM SIGMOD International Conference on Management of Data_ (pp. 1335-1349).

[5] Cvetkov-Iliev, A., Allauzen, A., & Varoquaux, G. (2023). Relational data embeddings for feature enrichment with background
information. _Machine Learning_, _112_(2), 687-720.

**Relation To Prior Work:**

The paper discusses well how it differs from existing RDB benchmarks. It does not provide a satisfactory evaluation of prior work on GNNs and alternative graph embedding methods.

**Summary And Contributions:**

The paper introduces 4DBInfer, a benchmark for GNN-based predictive modeling on RDBs, and discusses how to employ GNNs to RDBs to perform predictive tasks. The paper is supported by a robust online documentation and an extensive experimental section. However, it suffers from a poor presentation and poor discussion of prior work.

---

> ### Author Rebuttal · Authors · 2024-08-17
>
> We appreciate the detailed response and willingness to re-evaluate the score pending our rebuttal comments.  We also remark that the reviewer has listed three primary weaknesses (labeled W1, W2, and W3) in the "Opportunities For Improvement" section of Openreview.  These comments are restated and/or summarized in other sections as well (e.g., "Review," "Limitations," "Correctness," and "Clarity" etc.); however, to minimize the length of our response, we only respond once to each weakness.
>
> Before we begin though, we stress that our intent is not to favor GNNs, and in fact, we invested considerable effort/novelty in producing a dramatically more scalable version of DFS, such that it now can be significantly more efficient than GNNs as alluded to on Lines 340-342.  This is a notable non-GNN contribution in and of itself.
>
> **Comment:**
> *W1. [The submission] does not include any discussion on alternative methods for generating graph embeddings ([1], [2], [3]).*
>
> **Response:**
> These new references present embedding methods that apply to *homogeneous* graphs in strictly *transductive* settings; nor do they exploit node features when present.  In contrast, to best reflect real-world scenarios involving dynamic RDB data, our 4DBInfer benchmarks and baselines involve heterogeneous graphs (because of different table types), inductive learning tasks (e.g., because new rows and entities with missing target values can be added to an RDB), and tabular node features (when rows with rich attributes are converted to nodes).  And although there exist more recent graph embedding methods that extend to heterogeneous graphs (e.g., metapath2vec), these approaches are still not generally inductive and often require manually specifying dataset-specific metapaths.  At least in their present form then, they are not directly viable as automated/general-purpose baselines like the other models we have chosen.  The latter includes many non-GNN baselines formed via strong tabular base models combined with DFS or simple joins, all of which apply inductively and exploit row-wise features across multiple tables.
>
>
> **Comment:**
> *W1  ...  or prior work on converting tabular data into graph form ([4], [5]).*
>
> **Response:**
> Reference [4] the reviewer mentions presents a nice data integration approach and we are happy to cite it for additional context.  However, although it could potentially be combined with additional components for learning a predictive model (our focus), designing such new methodology represents a standalone research direction and is beyond our current scope of evaluating existing baselines.  Of course our 4DBInfer toolbox is set up such that others can easily explore such options in the future.
>
> Meanwhile reference [5] suggested by the reviewer (cited as [14] in our submission) does in fact present a complete pipeline called KEN for making predictions with relational data.  However, KEN is currently a transductive model (at least given the currently-available public codebase) and hence not applicable to most of our benchmarks.  Additionally, Table 1 in [5] shows that DFS + boosted trees, a baseline we already include in our submission, generally outperforms KEN in terms of prediction quality.
>
>
> **Comment:**
> *W1. If possible, I would strongly encourage the authors to try evaluate some of the simpler suggested baselines to make the submission more convincing...*
>
> **Response:**
> Per our comments above, the suggested baselines are not directly applicable in their present form.  We also emphasis that it was a *massive* undertaking to judiciously select, implement, streamline, and run the *many* current 4DBInfer baselines we already have, such that some omissions are inevitable.  Still, our hope is that additional models can be added, by ourselves and others, as research in this field progresses.
>
>
> **Comment:**
> *W2. There is no mention of the computational resources that were used to conduct the experiments (CPU/GPU/RAM usage) ... Please add a section on the resources required*
>
> **Response:**
> This is a great point, and easy to include.  For details, please see the rebuttal PDF uploaded to Openreview (near top of page). This includes a new table with timing comparisons and resource requirements for running 4DBInfer benchmarks, and definitely will be added to our revision.
>
> **Comment:**
> *W3. The paper is not well written...*
>
> **Response:**
> We agree that there is room to improve clarity and we will continue to do so (the reviewer's thoughtful suggestions for improvement are helpful in this regard).  We do though politely remark that writing quality evaluations are naturally subjective, and all other reviewers specifically mentioned that our paper is clear/well-written.
>
> **Comment:**
> *W3. The abstract begins by mentioning the prediction of missing column values in a target table (an imputation task), but from the body it is not clear ...*
>
> **Response:**
> Predicting missing values is indeed what is being evaluated in all benchmarks, noting that this includes missing values that may be present in new rows added to the RDB at test time, i.e., a form of inductive inference.  As for training, we simply mask out observable values in the target column of the training set. This is no different than single-table, tabular prediction tasks whereby known values in the target column are used as training labels.  See also Section 2.3 for the most general form of the training and inference specifications.
>
> **Comment:**
> *W3. In Figure 1, what does "billion-scale" mean?*
>
> **Response:**
> This refers to the number of rows in the RDB.  See Lines 79, 273, and 309-310 for context.
>
> **Comment:**
> *I was unable to install the package...*
>
> **Response:**
> We have provided a script to setup a conda environment with all the dependencies in our public repository; please see this [link](https://github.com/awslabs/multi-table-benchmark/blob/main/conda/create_conda_env.sh), and we are happy to assist further as needed.

---

> > ### Comment · Reviewer_RpkB · 2024-08-19
> >
> > I thank the authors for the response.
> >
> > Firstly, as I have experienced first hand the performance of the default implementation of DFS, I appreciate a lot the effort put into developing a more performant version of it, and I should have highlighted that better in the review.
> >
> > Regarding the baselines [1, 2, 3] and the arguments for why they don't work: I'm absolutely on board with the argument! The only issue I have is that it's not in the main body of the paper. Having examples of methods that _do not work_ (like these) would be a compelling argument for why something better than these references is needed, while not providing such examples looks like an omission (which is what I was not convinced by).
> >
> > Regarding [5]: I must have missed the [14], my mistake.
> >
> > On running the baselines: I am well acquainted with the effort required to run an additional baseline, especially when working with methods that are not easy to adapt to the scenario under observation. Clearly, while baselines would strengthen the paper, the previous discussion on why said baselines would not work would be sufficient to explain their absence.
> >
> > I appreciate the openness to feedback on the paper clarity and I agree that this point is strongly subjective. I subjectively maintain that the original version of the paper could be improved (as the two provided examples attest), and that if I raised the point in the review, then it is likely that other readers might have similar issues with the text. Admittedly, I could have been more cordial in how I raised the point.
> >
> > I believe the comments from the authors have addressed my comments satisfactorily, and I will raise my evaluation as a result.

---

> > ### Author Response · Authors · 2024-08-20
> > **Follow-Up to Reviewer RpkB**
> >
> > We sincerely appreciate the reassessment of our paper, as well as additional suggestions that continue to be constructive; our updated draft will surely be that much better upon hearing them.  And among other things, we agree that in our original draft it would have been helpful to provide more background regarding why we chose the baselines we did.  Indeed explicitly conveying our rationale thus far, and particularly why alternatives were *not* chosen as the reviewer has mentioned, plausibly increases the chances that readers will contribute even stronger baselines to 4DBInfer in the future.

---

> > ### Comment · Reviewer_RpkB · 2024-08-20
> >
> > Regarding the point of the resources required to work with 4DBInfer, I must say that they are quite steep, especially for practitioners with limited funding or access to such infrastructure. It is quite likely that having this kind of requirements will preclude researchers from running the benchmark, thus reducing the reach of this work.
> >
> > In the optic of future work, would it be feasible to produce a "distilled" version of this benchmark, to be run in lower-resource conditions? Or is it not possible to scale down the architecture?
> >
> > While I do not consider this issue as one that reduces the contribution, I do think that it is a limitation that should be acknowledged by the authors in the text, as it is in fact limiting the potential impact of the work.

---

> > ### Author Response · Authors · 2024-08-22
> > **Follow-up to Reviewer RpkB about resource requirements**
> >
> > In terms of lower resource environments, the central issue is often the size of the benchmarks themselves (which tend to be on the larger end to reflect real-world scenarios).  That being said, at least for some datasets, it is feasible to produce down-sampled temporal splits that do not undermine the legitimacy of the original tasks.  And there is precedent for this in the graph learning domain, e.g., the [Illinois Graph Benchmark](https://arxiv.org/abs/2302.13522) includes a graph dataset sampled to various different sizes.  We can certainly do this for larger 4DBInfer benchmarks as well (in fact for Outbrain we have already done so).  Good suggestion.
> >
> > We also point out that some of our current benchmarks are actually relatively small.  In particular, the two tasks we have defined using the Seznam dataset require only $\sim$ 1GB of memory for training, and can feasibly be handled on a newer laptop.

---

> > > ### Comment · Reviewer_RpkB · 2024-08-22
> > >
> > > Thanks for the response.
> > >
> > > > We also point out that some of our current benchmarks are actually relatively small. In particular, the two tasks we have defined using the Seznam dataset require only ~ 1GB of memory for training, and can feasibly be handled on a newer laptop.
> > >
> > > This does not come across from the text (or the provided addendum), as it is instead mentioned that "Most of the baselines provided by the 4DBInfer toolbox can run on an AWS g4dn.8xlarge instance...", with no specific detail on which are the "cheaper" tasks. Given that different tasks have such radically different footprints (as I can expect), it would be useful to have at least a ballpark measure of the resource requirements for each. For example, by adding peak RAM use to Table 2 in Appendix 2.
> > >
> > > As of now, I can imagine some readers may dismiss the benchmark out of hand as "all the tasks are too expensive to run", even if that isn't actually the case.

---

> > > > ### Author Response · Authors · 2024-08-22
> > > > **Follow-up to Reviewer RpkB about resource requirements**
> > > >
> > > > True, in our original response we only mentioned a loose minimum requirement for reproducing benchmarking results, not dataset-by-dataset requirements.  In any event, per the reviewer's follow-up suggestion, we can easily add a table with peak RAM usage stratified by each dataset.  Clearly this would better inform readers as to which benchmarks they can run on available hardware.  We will collect this information in the next day or so and update the paper accordingly.  Thanks for pointing out this issue.

---

> > > > > ### Comment · Reviewer_RpkB · 2024-08-22
> > > > >
> > > > > That would be useful! I have no further comments to make.

---

> > > > > > ### Author Response · Authors · 2024-09-01
> > > > > > **Follow-up to Reviewer RpkB about resource requirements**
> > > > > >
> > > > > > As key team members were traveling, there was some delay in collecting the requested RAM usage numbers; apologies for the delay.  In the table below, we have now included both peak GPU and CPU memory usage running 4DBInfer benchmarks using representative baseline methods.  For the DFS baseline, we use a 2-hop neighborhood and an MLP prediction head; meanwhile, for the R2N and R2N/E baselines we use an RGCN model.
> > > > > >
> > > > > > | **Dataset** | **Task** | **Baseline** | **Peak CPU Memory Usage (GB)** | **Peak GPU Memory Usage (GB)** |
> > > > > > |:-:|:-:|:-:|:-:|:-:|
> > > > > > | Amazon | Churn | DFS | 23.81 | 0.78 |
> > > > > > | | | R2N | 16.15 | 8.33 |
> > > > > > | | | R2N/E | 27.90 | 14.23 |
> > > > > > | | Purchase | DFS | 23.87 | 1.02 |
> > > > > > | | | R2N | 15.00 | 4.23 |
> > > > > > | | | R2N/E | 27.68 | 11.81 |
> > > > > > | | Rating | DFS | 23.79 | 1.01 |
> > > > > > | | | R2N | 15.27 | 13.00 |
> > > > > > | | | R2N/E | 27.04 | 13.80 |
> > > > > > | AVS | Repeater | DFS | 37.71 | 0.82 |
> > > > > > | | | R2N | 107.50 | 1.03 |
> > > > > > | | | R2N/E | 134.98 | 0.96 |
> > > > > > | Diginetica | CTR | DFS | 5.85 | 0.56 |
> > > > > > | | | R2N | 14.95 | 4.19 |
> > > > > > | | | R2N/E | 12.61 | 1.08 |
> > > > > > | | Purchase | DFS | 5.81 | 0.61 |
> > > > > > | | | R2N | 14.04 | 0.64 |
> > > > > > | | | R2N/E | 12.58 | 0.68 |
> > > > > > | MAG | Cite | DFS | 12.50 | 0.67 |
> > > > > > | | | R2N | 6.49 | 7.61 |
> > > > > > | | | R2N/E | 5.73 | 11.07 |
> > > > > > | | Venue | DFS | 9.93 | 0.80 |
> > > > > > | | | R2N | 5.17 | 4.63 |
> > > > > > | | | R2N/E | 4.24 | 7.48 |
> > > > > > | Outbrain-small | CTR | DFS | 1.37 | 1.01 |
> > > > > > | | | R2N | 2.31 | 3.04 |
> > > > > > | | | R2N/E | 2.08 | 13.74 |
> > > > > > | RetailRocket | CVR | DFS | 9.29 | 0.81 |
> > > > > > | | | R2N | 11.01 | 12.64 |
> > > > > > | | | R2N/E | 11.58 | 2.14 |
> > > > > > | Seznam | Charge | DFS | 1.54 | 0.84 |
> > > > > > | | | R2N | 2.01 | 12.32 |
> > > > > > | | | R2N/E | 2.08 | 3.84 |
> > > > > > | | Prepay | DFS | 1.59 | 0.79 |
> > > > > > | | | R2N | 2.06 | 7.55 |
> > > > > > | | | R2N/E | 2.11 | 12.10 |
> > > > > > | StackExchange | Churn | DFS | 15.87 | 0.93 |
> > > > > > | | | R2N | 5.74 | 13.33 |
> > > > > > | | | R2N/E | 11.77 | 3.04 |
> > > > > > | | Upvote | DFS | 15.93 | 2.30 |
> > > > > > | | | R2N | 4.71 | 4.56 |
> > > > > > | | | R2N/E | 10.45 | 6.11 |
> > > > > >
> > > > > > Note that except for Amazon and AVS, all experiments can be run on a modern gaming laptop such as [here](https://www.lenovo.com/us/en/p/laptops/legion-laptops/legion-5-series/lenovo-legion-5i-gen-9-(16-inch-intel)/83dgcto1wwus1).  For the largest benchmarks, the GPU memory usage can presently be covered by an NVIDIA RTX 3090, and can be easily reduced further by adopting a smaller batch size such as 128.

---

### Official Review · Reviewer_F71Q · 2024-07-23

**Rating:** 7
**Confidence:** 4
**Correctness:** The experiments and evaluation method…
**Clarity:** The paper is clearly written

**Review:**

I believe that the paper is of high quality and is very relevant to the community. It paves the way
for a novel research direction (of relational deep learning) with a high quality benchmark. The only
concern I have is with the baselines. Particularly, the lack of manual feature-engineering, which is
often an upper bound and an aproach to beat in real-world scenarios. I am willing to raise
the score if the "human-level" baseline performance aspect is addressed. For example there are open solutions to the acquire-valued-shoppers challenge ([1](https://github.com/MLWave/kaggle_acquire-valued-shoppers-challenge), [2](https://github.com/auduno/Kaggle-Acquire-Valued-Shoppers-Challenge/tree/master/features)). I believe such specialized solutions, which are often employed in real-world scenarious could beat the DFS baseline and the end-to-end methods present, this would be a usefull datapoint (even for a subset of datasets) and could provide a more clear "vector" for future progress. (see other less pressing concerns and suggestions bellow)

**Strengths:**

- Novel results in that end2end learning might be superior to the naive join baselines, and even automatic FE solutions
- A wide range of baseline methods and method design decisions are evaluated.
- Benchmark covers a wide range of tasks, and curates a significant amount of datasets
  - Paying attention to tasks real-world relevance
  - Broad range of graph-ML tasks covered
- Code is provided.
  - with faster DFS baseline
  - code quality is high, thus experimental results seem trustworthy
- The writing is clear and the paper structure is good.

**Additional Feedback:**

-

**Documentation:**

Dataset documentation is full and sufficient. The code would benefit from additional documentation (this would help build new research on it)

**Ethics:**

No ethical concerns

**Limitations:**

Limitations are adequately addressed.

**Opportunities For Improvement:**

**more important**:
- Addressing the human feature-engineering baseline solutions (strong FE + strong tabular model -- most ubiqutious case in many production real-world scenarios)
- Can you communicate the complexity and timings in the main text? The full trainig and full inference time comparisons for methods?
- Could you please provide more clear instructions on how to use the benchmark:
  - Current setup with all its degrees of freedom is a bit overwhelming. Are there tiers/tasks (just outline those in the text)?
  - Do we want methods to be tested with various graph extraction methods?
- Are graph selection/subsampling methods applied to the automatic FE solutions? Which subsampling
  methods are used, if any? Could you elaborate on this in the text?
- Do DFS/GNN solutions consider time-based leakage during dataset construction (e.g. do we use
  aggregations with future target values during the DFS feature creation for example)? I got the
  sense that this is covered in section 2.1, but could not find further concrete details.

**would be nice**:
- A leaderboard (e.g. a huggingface space with solutions like [OpenLLM
  leaderboard](https://huggingface.co/spaces/open-llm-leaderboard/open_llm_leaderboard) or upcoming
  [RelBench leaderboard](https://huggingface.co/spaces/relbench/leaderboard) would benefit both the
  benchmark adoption and the community.
- The `pip install dbinfer-bench` is not working (I believe it is not in pypi yet), however
  instructions for reproducing results work fine.
- Currently, iterating on new graph extraction or feature engineering methods is non-trivial in my
  opinion. The benchmark would benefit from documentation, more examples akin to the notebook
  present in the repository or simple standalone implementations of the core ideas (e.g a standalone
  data loading + dfs + XGBoost pipeline, graph dataset creation + GNN pipelines) -- something
  similar to https://github.com/vwxyzjn/cleanrl could greatly help new researchers in the field and
  may increase adoption.

**Relation To Prior Work:**

The prior work discussion is extensive and full.

**Summary And Contributions:**

This paper presents a benchmark for relational machine learning. It formalizes the process
of learning on relational tables and proposes a systematic evaluation framework. The framework
(called 4dbinfer) includes two schemes of DB to graph conversion and two kinds of predictive
architectures (end-to-end GNNs and feature-engineering with tabular ML architectures). Experiments
on the introduced benchmark suggest that there are no universally superior graph conversion methods
or model architectures. Furthermore, authors show promise of the end-to-end graph neural network
based solutions, which often outperform automatic FE (represented by deep feature synthesis). Overall, the
benchmark and accompanying experimental results are valuable contributions to the ML community.

---

> ### Author Rebuttal · Authors · 2024-08-17
>
> Thanks for pointing out the novelty of our results, significance of our curated datasets, and clarity of our writing.  We address each constructive comment provided by the reviewer in the "Review" and "Opportunities For Improvement" sections of Openreview.
>
>
> **Comment:**
> *The only concern I have is with the baselines. Particularly, the lack of manual feature-engineering, which is often an upper bound and an aproach to beat in real-world scenarios. I am willing to raise the score if the "human-level" baseline performance aspect is addressed... Addressing the human feature-engineering baseline solutions ...*
>
> **Response:**
> The tricky part with manual feature-engineering is that it is highly dependent on both the expertise of the data scientist and nuances of each dataset (unlike automated 4DBInfer pipelines that can be broadly applied by non-experts).  That being said, our simple join baselines represent common ingredients often found in manual solutions.  Moreover, in recent related work (reference [14] in our submission) several manual feature handcrafting baselines were evaluated against DFS+Tabular alternatives analogous to what we have included, with the latter performing significantly better (see Table 1 in [14] where DFS + gradient boosted trees generally outperforms other reported baselines).
>
> Of course admittedly stronger handcrafting is often possible per the reviewer's suggestion.  In this regard, during the rebuttal period we tried running the AVS challenge solutions (1,2) the reviewer mentioned; however, our AVS benchmark necessarily involves different splits from Kaggle (because of unknown competition test labels).  And at least on our AVS benchmark, solutions (1,2) did not operate successfully.  The first predicted all 0's after we generated the VowpaWabbit features, ran the VowpalWabbit command as specified, and generated test set predictions.  And the second solution crashed in the middle of feature generation, presumably for reasons related to the benchmark changes.  In any event, with more time we can perhaps get one or both methods to work, but these cases nonetheless highlight some of the challenges in leveraging handcrafted solutions within the development of a general-purpose benchmarking toolbox.
>
> Still, the reviewer's point is well-taken, and perhaps in the future we can conduct a user study to produce high-quality manual feature-engineering solutions for each 4DBInfer benchmark.  This would indeed be a useful calibration point for assessing the progress of automated methods as the reviewer has noted.
>
> **Comment:**
> *Can you communicate the complexity and timings in the main text? The full trainig and full inference time comparisons for methods?*
>
> **Response:**
> Some timing comparisons are included in Table 6; however, we have now produced additional evaluations in the new rebuttal PDF (link near top of page).  These will be added to the revision, and definitely improve the paper, good suggestion.
>
> **Comment:**
> *Could you please provide more clear instructions on how to use the benchmark:  (1) Current setup with all its degrees of freedom is a bit overwhelming. Are there tiers/tasks (just outline those in the text)? (2) Do we want methods to be tested with various graph extraction methods?*
>
> **Response:**
> While there are numerous potential use cases, one reasonable option for prototype model development would be to prioritize benchmarks that are most representative of an application of interest.  This is one reason to introduce a diverse set of tasks to begin with, increasing the chances that more real-world scenarios are covered.  And depending on model structure, we do believe that testing multiple graph extraction methods, and/or introducing new ones, can be beneficial.
>
>
> **Comment:**
> *Are graph selection/subsampling methods applied to the automatic FE solutions? Which subsampling methods are used, if any? Could you elaborate on this in the text?*
>
> **Response:**
> Good questions.  As discussed on Lines 204-210, DFS and simple joining methods can be derived as forms of subgraph sampling applied to graphs extracted via Row2Node.  Note that this sampling is designed to avoid temporal leakage by excluding nodes with future timesteps relative to a given target value.  While it is also possible to derive new variants of DFS based on other graph extraction procedures, we have not yet explored this possibility but hope to do so in the future.
>
>
>
> **Comment:**
> *Do DFS/GNN solutions consider time-based leakage during dataset construction (e.g. do we use aggregations with future target values during the DFS feature creation for example)? I got the sense that this is covered in section 2.1, but could not find further concrete details.*
>
> **Response:**
> All of our baseline models *exclude* future data when constructing input features.  For DFS in particular, please see Appendix F.1 for further details.  As for GNN baselines, we use temporal graph neighbor sampling (which strictly limits neighbors to those with earlier timestamps) implemented using the GraphBolt DGL package.
>
> **Comment:**
> *A leaderboard would benefit both the benchmark adoption and the community.*
>
> **Response:**
> Very true, and this can easily be added after the paper decision.
>
> **Comment:**
> *The pip install dbinfer-bench is not working (I believe it is not in pypi yet), however instructions for reproducing results work fine.*
>
> **Response:**
> Is the reviewer using Python 3?  On our side we observe no issue with the pip installation, and our project is available on PyPI as well at this [link](https://pypi.org/project/dbinfer-bench/).  Please let us know if there is a specific error message we could assist with.
>
> **Comment:**
> *Currently, iterating on new graph extraction or feature engineering methods is non-trivial in my opinion. The benchmark would benefit from documentation, more examples ...*
>
> **Response:**
> We agree, and more examples will incrementally be added and refined as part of regular 4DBInfer updates.

---

> > ### Comment · Reviewer_F71Q · 2024-08-19
> >
> > Thanks for the detailed response!
> >
> > - I've reconsidered my position regarding the necessity of the well tuned human-baseline. Thanks for the effort with trying to make those solutions work and mentioning the related work in [14] for studying human baselines. But please make sure to include the discussion regarding the possibility for producing high quality manual solutions (e.g. in the limitations and future work sections)
> > - Thanks for all the other clarifications
> > - The pip package installs fine as I checked now, that was probably an issue on my side
> > - I am encouraged that you mention your commitment to maintaining and updating the benchmark multiple time throughout the response
> >
> > I will increase the score to 7 if the discussion on human baselines would be added

---

> > > ### Author Response · Authors · 2024-08-20
> > > **Follow-Up to Reviewer F71Q**
> > >
> > > Thanks for considering our rebuttal points, and even trying out our toolbox (which we will continually try to improve moving forward).  And per the reviewer's suggestion, we can definitely include the discussion on human baselines when we update the draft.

---

> > > > ### Comment · Reviewer_F71Q · 2024-08-20
> > > >
> > > > Thanks! I updated my score

---

### Official Review · Reviewer_vDKj · 2024-07-25
**Review for 4DBInfer**

**Rating:** 6
**Confidence:** 3
**Correctness:** The claims in the submission are corr…
**Clarity:** The paper is well-written.

**Review:**

Quality: The experimental setup and clear, systematic development of both the benchmark datasets and the machine learning models. The authors also included diverse datasets and tasks to help evaluate the robustness of different predictive models.

Clarity: The paper is well-structured and clearly written.

Originality: The approach to integrating graph-centric modeling directly with RDBs and developing a dedicated benchmarking toolbox is novel.

Pros: Addressing the gap in benchmarks for predictive models on RDBs is a notable contribution, providing resources that could lead to more effective and optimized ML models for complex data structures.

Cons: It seems like biases could be introduced by the choice of graph conversion methods.

**Strengths:**

The development of a benchmarking toolbox for predictive models on relational databases is a novel contribution, given the increasing complexity of data in machine learning. The testing framework is also detailed and comprehensive.

**Additional Feedback:**

There's no additional feedback.

**Documentation:**

The work is well-documented.

**Ethics:**

There are no ethical concerns.

**Limitations:**

The method proposed is quite complex, which could gatekeep some potential readers/users.

**Opportunities For Improvement:**

There could be more varieties of data types, as well as additional evaluation metrics.

**Relation To Prior Work:**

The introduction includes its relation to prior work.

**Summary And Contributions:**

The paper introduces 4DBInfer, an open-source toolbox for benchmarking graph-centric predictive modeling on relational databases (RDBs). The main contribution is evaluating machine learning models that operate directly on RDBs by providing robust benchmarks and baseline models. The toolbox facilitates the transformation of multi-table datasets into graphs, preserving tabular characteristics, and supports various predictive models tuned to handle these structures. Additionally, the paper details the creation of a diverse collection of large-scale RDB datasets and associated predictive tasks, enhancing the realism and applicability of model evaluations.

---

> ### Author Rebuttal · Authors · 2024-08-17
>
> Thanks for pointing out our clear, systematic development, novel toolbox, and well-written paper.  We address the single con and limitation raised by the reviewer as follows.
>
> **Comment:**
> *Cons: It seems like biases could be introduced by the choice of graph conversion methods.*
>
> **Response:**
> Absolutely, biases can be introduced by graph conversion methods.  However, we do not view this as a con of 4DBInfer, but rather, *a notable strength*. In particular, unlike prior work, we explicitly highlight the importance of considering different graph construction methods, and have specifically designed 4DBInfer to handle multiple approaches while leaving the door open for future alternatives.  In this way biases can be carefully scrutinized, and where appropriate, exploited to improve predictive performance.
>
> **Comment:**
> *The method proposed is quite complex, which could gatekeep some potential readers/users.*
>
> **Response:**
> Building predictive models of relational data is by its very nature a complex undertaking.  While this is reflected in our proposed 4DBInfer toolbox, it is our ultimate hope that this release will actually lower the barrier to entry, even if certain complexities are infeasible to remove entirely.

---

> > ### Comment · Reviewer_vDKj · 2024-08-31
> >
> > Thank you for addressing my concerns. I will keep my score.

---

### Official Review · Reviewer_p2fN · 2024-07-25
**Good work with well-documented library**

**Rating:** 7
**Confidence:** 4

**Review:**

1. The paper is well written with through experimentation, ablation studies and rigorous analysis of tasks, datasets and implementation.
2. The github project is well documented with a clean API, getting started tutorials and documentation.
3. The problem is well motivated and the approach is novel for RDB benchmarking by combining multiple dimensions of analysis.
4. The benchmarking protocol integrates diverse approaches into a unified framework, but which independently may not be entirely new.
5. The work has potential to drive significant impact in evaluation of ML methods on large datasets (which require an RDB to express).

I believe this is a through and rigorous work, with clean paper and a well documented github repo, which deserves to be accepted.

**Strengths:**

1. Contribution:
- Introduces 4DBInfer, a comprehensive benchmarking toolbox for predictive modeling on relational databases (RDBs)
- Addresses a critical gap in existing benchmarks by focusing on raw RDB data rather than pre-processed graphs
- Provides a unified framework for evaluating diverse modeling approaches across multiple dimensions
2. Analysis:
- Covers a wide range of datasets, tasks, graph extraction methods, and predictive models.
- Includes both graph-based and tabular approaches, allowing for comprehensive comparisons.
- Scales up to billion-row datasets, addressing real-world big data challenges
3. Github:
- Authors release an open-source toolbox that can be readily used by other researchers.
- The modular design allows for easy integration of new datasets, tasks, or models.
- Also provide a standardized way to compare different approaches, potentially reducing siloed research

**Additional Feedback:**

I would encouraging adding more utility examples in the github repo and adding some kind of a benchmarking leaderboard for standard datasets. That can help people quickly appreciate the impact of your work and also provide some relevance to your efforts.

**Clarity:**

The paper is well written with clear flow of thoughts, extensive experiments, good discussion and rigorous ablation studies. I al

**Correctness:**

Yes, the claims are well documented with thorough experiments, ablation studies and discussion. My only concern remains regarding the quality of baselines. Would be good for authors to do more analysis there

**Documentation:**

The github repo is well documented with a clear API and good documentation and tutorials.

**Ethics:**

No, there are no concerns.

**Limitations:**

Addressed limitations:

1. The authors acknowledge that their baseline models may not be the absolute state-of-the-art in each domain.
2. They discuss the challenges of fairly comparing diverse approaches (e.g., tabular vs. graph-based methods).
3. They mention the need for ongoing maintenance and updates to keep the benchmark relevant.

Potential improvements regarding limitations:

1. Include a dedicated section on limitations and future work to more explicitly address potential shortcomings of the approach.
2. Discuss the computational resources required to run the full benchmark, as this could be a limitation for some researchers.
3. Address potential biases in the dataset selection and task formulation process.

**Opportunities For Improvement:**

1. The complexity of the 4D framework might be overwhelming for some practitioners, potentially limiting adoption. Would be helpful to build open--benchmarks on datasets community cares about today.
2. Some baseline models may not represent the absolute state-of-the-art in their respective domains
3. Limited discussion of statistical significance of results, which could affect the reliability of comparisons
4. The paper could benefit from more rigorous theoretical analysis of the proposed graph extraction methods
5. The paper could benefit from more ablation studies to isolate the impact of each dimension in the 4D framework

**Relation To Prior Work:**

Yes, this is discussed in paper but they acknowledge baselines may not be state-of-the-art.

**Summary And Contributions:**

This submission introduces 4DBInfer, a new benchmarking toolbox for evaluating predictive modeling approaches on relational databases (RDBs). The key contributions are:

1. A 2D space of baseline models, including various graph construction/sampling methods and predictive architectures spanning both graph neural networks and tabular models.
2. A 2D suite of RDB benchmarking datasets and tasks, covering diverse domains, scales (up to 2B rows), schemas, and temporal structures.
3. The 4DBInfer toolbox itself, which operationalizes evaluation across the 4 dimensions of datasets, tasks, graph extractors, and predictive models.
4. Empirical results using 4DBInfer that demonstrate the importance of considering all 4 dimensions when developing RDB predictive models.
5. A clean API with documentation, installation guide and tutorials

The authors argue that 4DBInfer addresses limitations of existing benchmarks by:
1. Using raw RDB data rather than pre-processed graphs
2. Including larger-scale datasets
3. Covering more diverse domains and tasks
4. Enabling comparisons across different graph extraction methods
5. Facilitating evaluation of both graph and tabular models

---

> ### Author Rebuttal · Authors · 2024-08-17
>
> Thanks for the summary of our work as thorough and rigorous, with a clean paper and well-documented github repo, that deserves to be accepted.  We also thoroughly appreciate the reviewer's detailed suggestions for improvement, which we respond to as follows.
>
> **Comment:**
> *1. The complexity of the 4D framework might be overwhelming for some practitioners, potentially limiting adoption. Would be helpful to build open--benchmarks on datasets community cares about today.*
>
> **Response:**
> For broad applicability and relevance, our current design covers wide-ranging scenarios and hence by its very nature is complex.  And we do encourage the community to add additional datasets of interest to 4DBInfer to increase coverage of impactful domains (although some popular candidates unfortunately have licensing restrictions that, at least for now, prevent us from adding them).
>
> **Comment:**
> *2. Some baseline models may not represent the absolute state-of-the-art in their respective domains.*
>
> **Response:**
> Indeed there may be existing approaches that outperform our current selection of baselines, and it is our hope that practitioners will experiment with such methods to push the envelop further, facilitated by the neutral, standardized setting provided by 4DBInfer.  And if the reviewer has additional suggestions to include, we would surely benefit from them.
>
> **Comment:**
> *3. Limited discussion of statistical significance of results, which could affect the reliability of comparisons*
>
> **Response:**
> Good suggestion.  As it turns out, the trial-to-trial variability is quite small for the benchmarks we have chosen.  For a representative sample, please see the new table below with standard errors included across 5 trials.  As can be observed from these results, variability is limited to the 3rd or 4th significant digit, which has little impact on performance comparisons.  We can include these results in the revision.
>
> | Dataset/Task | Metric | DFS/MLP | R2N/RGCN |
> |:-------:|:-------:|:-------:|:------:|
> | Amazon/Rating | RMSE↓ | 0.9845 ± 0.0008 | 0.9640 ± 0.0016 |
> | Outbrain/CTR  | AUC↑ | 0.5503 ± 0.0012 | 0.6239 ± 0.0026 |
> | Seznam/Charge | Acc↑ | 0.7518 ± 0.0027 | 0.7902 ± 0.0012 |
>
>
> **Comment:**
> *4. The paper could benefit from more rigorous theoretical analysis of the proposed graph extraction methods.*
>
> **Response:**
> We completely agree that there are many research directions related to the theoretical analysis of graph extraction methods.  We provide one line of preliminary analysis in Appendix I.3, but there are many others.
>
> **Comment:**
> *5. The paper could benefit from more ablation studies to isolate the impact of each dimension in the 4D framework.*
>
> **Response:**
> Appendices A and J contain what amounts to an extensive series of ablations.  However, given the complexity of our 4D design space, we agree that more exploration would likely expose new insights.
>
> **Comment:**
> *Potential improvements regarding limitations: (1) Include a dedicated section on limitations and future work to more explicitly address potential shortcomings of the approach. (2) Discuss the computational resources required to run the full benchmark, as this could be a limitation for some researchers. (3) Address potential biases in the dataset selection and task formulation process.*
>
> Great suggestions for a revision.  And as for point (2) in particular, please see the  rebuttal PDF uploaded to Openreview (near top of page).  This includes a new table with timing comparisons and resource requirements for running 4DBInfer benchmarks.

---

### Author Rebuttal · Authors · 2024-08-17

In additional to reviewer-specific rebuttals below, we also include here a PDF with new timing comparisons and details of resource requirements for running all 4DBInfer benchmarks.

---

### Decision · Program_Chairs · 2024-09-26

**Decision:**

Accept (Poster)

**Comment:**

This paper addresses the task of predictive modeling for relational databases. The authors explore existing baselines for these tasks, and assemble a collection of RDB-based benchmark tasks. Reviewers agree the paper is well written and thorough, the benchmark itself has a clean API and documentation, and provides an important, novel contribution to the community, particularly a benchmark that is intended for learning and evaluating on relational tables. While some limitations remain and can be addressed by future work, overall we concur with the reviewers that this paper clears the bar for publication at NeurIPS Datasets and Benchmarks.